# Unsupervised Visual Representation Learning via Mutual Information Regularized Assignment

**Dong Hoon Lee**
KAIST*
donghoonlee@kaist.ac.kr

**Sungik Choi**
LG AI Research
sungik.choi@lgresearch.ai

**Hyunwoo Kim**
LG AI Research
hwkim@lgresearch.ai

**Sae-Young Chung**
KAIST
sychungster@naver.com

## Abstract

This paper proposes Mutual Information Regularized Assignment (MIRA), a pseudo-labeling algorithm for unsupervised representation learning inspired by information maximization. We formulate online pseudo-labeling as an optimization problem to find pseudo-labels that maximize the mutual information between the label and data while being close to a given model probability. We derive a fixed-point iteration method and prove its convergence to the optimal solution. In contrast to baselines, MIRA combined with pseudo-label prediction enables a simple yet effective clustering-based representation learning without incorporating extra training techniques or artificial constraints such as sampling strategy, equipartition constraints, etc. With relatively small training epochs, representation learned by MIRA achieves state-of-the-art performance on various downstream tasks, including the linear/$k$-NN evaluation and transfer learning. Especially, with only 400 epochs, our method applied to ImageNet dataset with ResNet-50 architecture achieves 75.6% linear evaluation accuracy. Our implementation is available at `https://github.com/movinghoon/mira`.

## 1   Introduction

There has been a growing interest in using a large-scale dataset to build powerful machine learning models [45]. Self-supervised learning (SSL), which aims to learn a useful representation without labels, is suitable for this trend; it is actively studied in the fields of natural language processing [19, 20] and computer vision [10, 30]. In the vision domain, recent SSL methods commonly use data augmentations and induce their visual representation to be augmentation-invariant. They have achieved state-of-the-art performance surpassing supervised representation in a variety of visual tasks, including semi-supervised learning [8, 53], transfer learning [22], and object detection [13].

Meanwhile, a line of work uses clustering for un-/self-supervised representation learning. They explicitly assign pseudo-labels to embedded representation via clustering, and the model is thereby trained to predict such labels. These clustering-based methods can account for inter-data similarity; representations are encouraged to encode the semantic structure of data. Prior works [51, 49, 4, 32] have shown encouraging results in small-scaled settings; Caron et al. [6] show that it can also be applied to the large-scaled dataset or even to a non-curated dataset [7]. Recently, several works [2, 8, 39] have adopted the philosophy of augmentation invariance and achieved strong empirical results.

---

*Work partially done during internship at LG AI Research.

36th Conference on Neural Information Processing Systems (NeurIPS 2022).

They typically assign pseudo-labels using augmented views while predicting the labels by looking at other differently augmented views.

Despite its conceptual simplicity, a naive application of clustering to representation learning is hard to achieve, especially when training with large-scale datasets. This is because clustering-based methods are prone to collapse, i.e., all samples are assigned to a single cluster; hence, recent methods heavily rely on extra training techniques or artificial constraints, such as pre-training [50], sampling strategy [6], equipartition constraints [2, 8], to avoid collapsing. However, it is unclear if these additions are appropriate or how such components will affect the representation quality.

In this paper, we propose Mutual Information Regularized Assignment (MIRA), a pseudo-labeling algorithm that enables clustering-based SSL without any artificial constraints or extra training techniques. MIRA is designed to follow the infomax principle [40] and the intuition that *good labels are something that can reduce most of the uncertainty about the data*. Our method assigns a pseudo-label in a principled way by constructing an optimization problem. For a given training model that predicts pseudo-labels, the optimization problem finds a solution that maximizes the mutual information (MI) between the pseudo-labels and data while considering the model probability. We formulate the problem as a convex optimization problem and derive the necessary and sufficient condition of solution with the Karush-Kuhn-Tucker (KKT) condition. This solution can be achieved by fixed-point iteration that we prove the convergence. We remark that MIRA does not require any form of extra training techniques or artificial constraints, e.g., equipartition constraints.

We apply MIRA to clustering-based representation learning and verify the representation quality on several standard self-supervised learning benchmarks. We demonstrate its state-of-the-art performance on linear/$k$-NN evaluation, semi-supervised learning, and transfer learning benchmark. We further experiment with convergence speed, scalability, and different components of our method.

Our contributions are summarized as follows:

- We propose MIRA, a simple and principled pseudo-label assignment algorithm based on mutual information. Our method does not require extra training techniques or artificial constraints.

- We apply MIRA to clustering-based representation learning, showing comparable performance against the state-of-the-art methods with half of the training epochs. Specifically, MIRA achieves 75.6% top-1 accuracy on ImageNet linear evaluation with only 400 epochs of training and the best performance in 9 out of 11 datasets in transfer learning.

- Representation by MIRA also consistently improves over other information-based SSL methods. Especially our method without multi-crop augmentation achieves 74.1% top-1 accuracy and outperforms BarlowTwins [53], a baseline information maximization-based self-supervised method.

## 2 Related works

**Self-supervised learning** SSL methods are designed to learn the representation by solving pretext tasks, and recent state-of-the-art methods encourage their learned representations to be augmentation invariant. They are based on various pretext tasks: instance discrimination [10, 11, 13, 14], metric learning [28, 12], self-training [54, 9], and clustering [2, 6, 8]; only a few account for encoding the semantic structure of data. While some works [47, 21, 35] consider the nearest neighbors in the latent space, our method belongs to the clustering-based SSL method that flexibly accounts for inter-data similarity. Meanwhile, many SSL methods are prone to collapsing into a trivial solution where every representation is mapped into a constant vector. Various schemes and mechanisms are suggested to address this, e.g., the asymmetric structure, redundancy reduction, etc. We will review more relevant works in detail below.

**Collapse preventing** Many SSL approaches rely on extra training techniques and artificial assumptions to prevent collapsing. In clustering-based methods, DeepCluster [6] adapts a sampling strategy to sample elements uniformly across pseudo-labels to deal with empty clusters; SeLa [2] and SwAV [8] impose equipartition constraints to balance the cluster distribution. Similarly, SelfClassifier [1] uses a uniform pseudo-label prior, and PCL [39] employs concentration scaling. DINO [9] and ReSSL [54] address collapsing by specific combinations of implementation details, i.e., centering and scaling with an exponential moving average network; their mechanism for preventing collapse

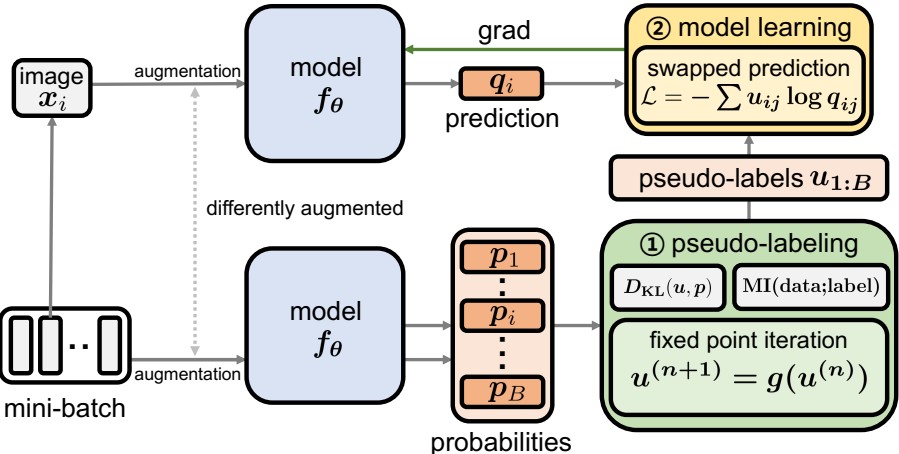

Figure 1: **Overview of representation learning via MIRA.** In our representation learning, MIRA provides pseudo-labels with model probabilities, and the model is learned by predicting the pseudo-labels. Our main contribution is in the ① **pseudo-labeling** process that accounts for mutual information between the pseudo-label and data. In MIRA, optimal pseudo-labels are computed through the fixed-point iteration (Eq. 7). Given such pseudo-labels, ② **model updates** its parameters by gradient update on swapped prediction loss.

is unclear. In this work, we show our method can naturally avoid collapsing without any of these assumptions or training techniques. We achieve results better than baselines with a simple but novel information regularization algorithm. We take a more detailed comparison with SeLa and SwAV after explaining our method in Sec. 3.3.

**Information maximization**    Information maximization is a principal approach to learn representation and to avoid collapse. DeepInfoMax [31] propose the MI maximization between the local and global views for representation learning; the existence of negative pairs prevents training toward the trivial solution. BarlowTwins [53] and W-MSE [23] address the collapsing with redundancy reduction that indirectly maximizes the content information of embedding vectors. Among clustering-based approaches, IIC [34] maximizes the MI between the embedding codes to enable representation learning; similar to ours, TWIST [26] proposes combining the MI between the data and class prediction as a negative loss term with an augmentation invariance consistency loss. Both IIC and TWIST use the MI as a loss function and directly optimize their model parameters with gradient descent of the loss. However, the direct optimization of MI terms by updating model parameters often leads to a sub-optimal solution [26]; TWIST copes with this issue by appending the normalization layer before softmax and introducing an additional self-labeling stage. In contrast, MIRA addresses the difficulty of MI maximization in a principled way via explicit optimization.

## 3    Method

In this section, we explain our pseudo-labeling algorithm–MIRA. When applying MIRA to representation learning, we follow the basic framework of clustering-based representation learning that alternates between *pseudo-labeling*, i.e., cluster assignments, and *model training* to predict such labels. Figure 1 illustrates our representation training cycle. We will first explain our main contribution, MIRA (*pseudo-labeling*) and then explain how it applies to *model training*.

Our idea is to employ the information maximization principle into pseudo-labeling. We formulate an optimization problem for online clustering that assigns soft pseudo-labels to mini-batch samples (Sec. 3.1). The problem minimizes the KL divergence between the model prediction (probability) and pseudo-label while maximizing the mutual information between the data and pseudo-label. We propose an iterative fixed point method to solve the optimization problem (Sec. 3.2). For the model training, we use the swapped prediction loss [8] (Sec. 3.3).

## 3.1 Mutual information regularized pseudo-labeling

We have a classification model[2] $f_\theta$ parametrized by $\theta$ that outputs $K$-dimensional logit $f_\theta(\boldsymbol{x}) \in \mathbb{R}^K$ for an image $\boldsymbol{x}$, where $K$ is a predefined number of clusters. The model probability $\boldsymbol{p}$ of an image $\boldsymbol{x}$ is then given by the temperature $\tau_t$ scaled output of the model—$\boldsymbol{p} := \text{softmax}(f_\theta(\boldsymbol{x})/\tau_t)$—as in Caron et al. [8, 9]. For a mini-batch of input images $\boldsymbol{X} = \{\boldsymbol{x}_i\}_{i=1}^B$, we denote the set of model probabilities $\boldsymbol{P} = \{\boldsymbol{p}_i\}_{i=1}^B \subset \mathbb{R}^K$. In our pseudo-labeling, for the given model probabilities $\boldsymbol{P}$, we want to assign pseudo-labels $\boldsymbol{W^*} = \{\boldsymbol{w}_i^*\}_{i=1}^B$ that will be used for training the model by predicting them.

We argue that such pseudo-labels should maximize the mutual information (MI) between themselves and data while accounting for the model probabilities $\boldsymbol{P}$. Let $\mathcal{B} \in \{1, ..., B\}$ and $\mathcal{Y}_{\boldsymbol{W}} \in \{1, ..., K\}$ be the random variables associated with the data index in mini-batch and labels by probability distributions $\boldsymbol{W} = \{\boldsymbol{w}_i\}_{i=1}^B$, respectively. Our online pseudo-label (cluster) assignment is determined by solving the following optimization problem:

$$\boldsymbol{W^*} = \arg\min_{\boldsymbol{W} \subset \Delta_K} \frac{1}{B} \sum_{i=1}^B D_{\text{KL}}(\boldsymbol{w}_i, \boldsymbol{p}_i) - \beta \hat{I}(\mathcal{Y}_{\boldsymbol{W}}; \mathcal{B}), \tag{1}$$

where $\Delta_K := \{\boldsymbol{w} \in \mathbb{R}_+^K \mid \boldsymbol{w}^\intercal \mathbf{1}_K = 1\}$, $\hat{I}$ indicates an empirical (Monte Carlo) estimates of MI, and $\beta$ is a trade-off parameter. The problem consists of the (1) KL divergence term that makes pseudo-labels to be based on the model probability $\boldsymbol{p}$ and (2) MI term between the pseudo-labels and data to induce more information about data into the pseudo-labels. By combining these two terms, we provide a refined pseudo-label that take account of both the model probability and MI.

To make the optimization problem tractable, we substitute the MI term $\hat{I}$ with the mini-batch estimates of the entropy $\hat{H}(\mathcal{Y}_{\boldsymbol{W}}|\mathcal{B})$ and marginal entropy $\hat{H}(\mathcal{Y}_{\boldsymbol{W}})$ in Eq. 2. We get:

$$\hat{I}(\mathcal{Y}_{\boldsymbol{W}}; \mathcal{B}) = \hat{H}(\mathcal{Y}_{\boldsymbol{W}}) - \hat{H}(\mathcal{Y}_{\boldsymbol{W}}|\mathcal{B}) = -\sum_{j=1}^K \bar{w}_j \log \bar{w}_j + \frac{1}{B} \sum_{i=1}^B \sum_{j=1}^K w_{ij} \log w_{ij}, \tag{2}$$

$$\frac{1}{B} \sum_{i=1}^B D_{\text{KL}}(\boldsymbol{w}_i, \boldsymbol{p}_i) = -\frac{1}{B} \sum_{i=1}^B \sum_{j=1}^K w_{ij} \log p_{ij} + \frac{1}{B} \sum_{i=1}^B \sum_{j=1}^K w_{ij} \log w_{ij}, \tag{3}$$

$$\boldsymbol{W^*} = \arg\min_{\boldsymbol{W} \subset \Delta_K} -\frac{1}{B} \sum_{i=1}^B \sum_{j=1}^K w_{ij} \log p_{ij} + \frac{1-\beta}{B} \sum_{i=1}^B \sum_{j=1}^K w_{ij} \log w_{ij} + \beta \sum_{j=1}^K \overline{w}_j \log \overline{w}_j, \tag{4}$$

where $\overline{w}_j = \frac{1}{B} \sum_{i=1}^B w_{ij}$ is the marginal probability of a cluster $j$ with $\boldsymbol{W}$. In practice, we find the optimal point $\boldsymbol{W^*}$ of the optimization problem Eq. 4 for pseudo-labeling.

## 3.2 Solving strategy

To solve efficiently, we propose a fixed point iteration that guarantees convergence to the unique optimal solution $\boldsymbol{W^*}$ of our optimization problem. The method is based on the following proposition.

**Proposition 1.** *For $\beta \in [0, 1)$, the problem Eq. 4 is a strictly convex optimization problem with a unique optimal point $\boldsymbol{W^*}$ that satisfies the following necessary and sufficient condition:*

$$\forall (i,j) \in \{1, ..., B\} \times \{1, ..., K\}, \quad w^*_{ij} = \frac{\overline{w^*}_j^{-\frac{\beta}{1-\beta}} p_{ij}^{\frac{1}{1-\beta}}}{\sum_{k=1}^K \overline{w^*}_k^{-\frac{\beta}{1-\beta}} p_{ik}^{\frac{1}{1-\beta}}}. \tag{5}$$

The proposition 1 is driven by applying the Karush-Kuhn-Tucker (KKT) conditions to the optimization problem Eq. 4. By substituting the necessary and sufficient condition (Eq. 5) into the marginal probability $\overline{w}_j = \frac{1}{B} \sum_{i=1}^B w_{ij}$, we get the necessary and sufficient condition for $\overline{w^*}$:

$$\overline{w^*}_j = \overline{w^*}_j^{-\frac{\beta}{1-\beta}} \frac{1}{B} \sum_{i=1}^B \frac{p_{ij}^{\frac{1}{1-\beta}}}{\sum_{k=1}^K \overline{w^*}_k^{-\frac{\beta}{1-\beta}} p_{ik}^{\frac{1}{1-\beta}}} \Leftrightarrow \overline{w^*}_j = \left[ \frac{1}{B} \sum_{i=1}^B \frac{p_{ij}^{\frac{1}{1-\beta}}}{\sum_{k=1}^K \overline{w^*}_k^{-\frac{\beta}{1-\beta}} p_{ik}^{\frac{1}{1-\beta}}} \right]^{1-\beta}. \tag{6}$$

---

[2]In our setting, the model consists of an encoder, projection head, and classification (prototype) head as in Caron et al. [8, 9]; the encoder output will be used as a representation.

Based on Eq. 6, we propose the update rule for $\{u_j^{(n)}\}_{j=1}^K \subset \mathbb{R}_+$ using the fixed point iteration as follows:

$$\forall j \in \{1, ..., K\}, \quad u_j^{(n+1)} = \left[ \frac{1}{B} \sum_{i=1}^B \frac{p_{ij}^{\frac{1}{1-\beta}}}{\sum_{k=1}^K (u_k^{(n)})^{-\frac{\beta}{1-\beta}} p_{ik}^{\frac{1}{1-\beta}}} \right]^{1-\beta}, \tag{7}$$

where $u_j^{(n)}$ converges to $\overline{w^*}_j$ as $n \to \infty$ since the necessary and sufficient condition (Eq. 6) is satisfied at the convergence. We can easily get $w^*_{ij}$ by Eq. 5 when the optimal marginal probability $\overline{w^*}_j$ is given. The proof of the proposition and convergence is in the Appendix.

By using the iterative updates of Eq. 7, we get our desirable pseudo-labels $\boldsymbol{W}^*$. This requires a few lines of code that are simple to implement. We observe that a few steps of iteration are enough for training. This is supported by the convergence analysis in Sec. 4.3. We use this fixed point iteration for pseudo-labeling and name the method–**M**utual **I**nformation **R**egularized **A**ssignment (**MIRA**) since it finds the pseudo-labels that are regularized by the mutual information.

### 3.3 Representation learning with MIRA

We explain how our pseudo-labeling method is applied to representation learning. We integrate the computed pseudo-labels with swapped prediction loss [8] to train the model; the model is trained to predict the pseudo-labels from the augmented views using the other views. Specifically, given two mini-batches of differently augmented views $\boldsymbol{X}^{(m)} = \{\boldsymbol{x}_i^{(m)}\}_{i=1}^B, m \in \{1, 2\}$, MIRA independently assigns pseudo-labels $\boldsymbol{U}^{(m)} = \{\boldsymbol{u}_i^{(m)}\}_{i=1}^B$ for each mini-batch. In parallel, model $f_\theta$ provides the temperature $\tau_s$ scaled softmax predictions[3] $\boldsymbol{Q}^{(m)} = \{\boldsymbol{q}_i^{(m)}\}_{i=1}^B$ of each mini-batch. The swapped prediction loss is given as follows:

$$L(\boldsymbol{X}^{(1)}, \boldsymbol{X}^{(2)}) = \ell(\boldsymbol{U}^{(1)}, \boldsymbol{Q}^{(2)}) + \ell(\boldsymbol{U}^{(2)}, \boldsymbol{Q}^{(1)})$$

$$= -\frac{1}{B} \sum_{i=1}^B \sum_{j=1}^K u_{ij}^{(1)} \log q_{ij}^{(2)} - \frac{1}{B} \sum_{i=1}^B \sum_{j=1}^K u_{ij}^{(2)} \log q_{ij}^{(1)}. \tag{8}$$

This loss function (Eq. 8) is minimized with respect to the parameters $\theta$ of the model $f_\theta$ used to produce the predictions $\boldsymbol{Q}^{(1)}, \boldsymbol{Q}^{(2)}$. For more detailed information about swapped prediction loss, please refer to Caron et al. [8].

The pseudo-code of MIRA for representation learning with Eq. 8 is provided in the Appendix. In the following experiments, we verify the effectiveness of MIRA for a representation learning purpose. We note that MIRA can integrate recently suggested self-supervised learning components, such as exponential moving average (EMA) or multi-crop (MC) augmentation strategy following the baselines [14, 8, 9]. For convenience, in the rest of this paper, we call the *representation learning with MIRA* also as MIRA. We discuss some further details as follows:

**Preventing collapse** The MI term in Eq. 4 takes a minimum value when collapsing happens. MIRA naturally avoids collapsed solution via penalizing assignment that exhibits low MI. Specifically, unless starting from the collapsed state, MIRA finds MI-maximizing points around the model prediction; it will not choose collapsed pseudo-labels. Hence, the iterative training to predict such labels will not collapse whenever the prediction of pseudo-labels is achievable. Our empirical results verify that MIRA does not require extra training techniques or artificial constraints to address collapsing.

**Comparison to SwAV and SeLa** SeLa [2] and SwAV [8] formulate their pseudo-labeling process into optimization problems, i.e., optimal transport (OT) problem, and solve it iteratively with Sinkhorn-Knopp (SK) algorithm [16]. To avoid collapse and apply the SK algorithm, they assume the equipartition of data into clusters. Mathematically, the difference to MIRA is in *how to deal with the marginal entropy*. SeLa and SwAV constrain the marginal entropy to maximum value–equipartition while MIRA decides it by MI regularization[4]. Asano et al. [2] argue that their pseudo-labels with the OT problem maximize the MI between labels and data indices under the equipartition constraint.

---

[3]That is $\boldsymbol{q}_i^{(m)} := \mathrm{softmax}(f_\theta(\boldsymbol{x}_i^{(m)})/\tau_s)$.

[4]Adding the equipartition constraint into Eq. 4, our problem converts to the OT problem of SwAV [8].

Table 1: **Linear evaluation with respect to training epochs.** All models use a ResNet-50 encoder and trained on training set of ImageNet. † are results from [12]. Results style: **best**, second best

| Method | Epochs 100 | 200 | 400 | 800 |
|---|---|---|---|---|
| *without multi-crop augmentations* | | | | |
| SimCLR† [10] | 66.5 | 68.3 | 69.8 | 70.4 |
| BYOL† [28] | 66.5 | 70.6 | 73.2 | **74.3** |
| SimSiam† [30] | 68.1 | 70.0 | 70.8 | 71.3 |
| MoCo-v3 [14] | 68.9 | - | - | 73.8 |
| DeepCluster-v2 [8] | - | - | 70.2 | - |
| SwAV† [8] | 66.5 | 69.1 | 70.7 | 71.8 |
| TWIST [26] | **70.4** | 70.9 | 71.8 | 72.6 |
| MIRA | 69.4 | 72.3 | 73.3 | 74.1 |
| *with multi-crop augmentations* | | | | |
| DeepCluster-v2 [8] | - | - | - | 75.2 |
| SwAV [8] | 72.1 | 73.9 | 74.6 | 75.3 |
| TWIST [26] | 72.9 | 73.7 | 74.4 | 74.1 |
| MIRA | **73.6** | **74.9** | **75.6** | - |

Table 2: **Linear evaluation on ImageNet.** Comparison with other self-supervised methods on ImageNet. *SL* denotes for *self-labeling* by [26]. Results style: **best**

| Method | Arch. | Epochs | Top-1 | Top-5 |
|---|---|---|---|---|
| Supervised | R50 | - | - | - |
| PCL [39] | R50 | 200 | 67.6 | - |
| SimSiam [12] | R50 | 800 | 71.3 | - |
| SimCLR-v2 [11] | R50 | 800 | 71.7 | - |
| InfoMin [46] | R50 | 800 | 73 | 91.1 |
| BarlowTwins [53] | R50 | 1000 | 73.2 | 91.0 |
| VicReg [3] | R50 | 1000 | 73.2 | 91.1 |
| SelfClassifier [1] | R50 | 800 | 74.1 | - |
| TWIST w/o *SL* [26] | R50 | 800 | 74.1 | - |
| BYOL [28] | R50 | 1000 | 74.3 | 91.6 |
| MoCo-v3 [14] | R50 | 1000 | 74.6 | - |
| DeepCluster-v2 [8] | R50 | 800 | 75.2 | - |
| SwAV [8] | R50 | 800 | 75.3 | - |
| DINO [8] | R50 | 800 | 75.3 | - |
| TWIST w/ *SL* [26] | R50 | 450 | 75.5 | - |
| MIRA | R50 | 400 | **75.6** | **92.5** |

However, it more resembles assuming MI maximization and then finding the cluster assignments that are optimal transport to the model prediction. In contrast, MIRA directly maximizes the MI by regularization without artificial constraints. While SwAV performs better than SeLa in most self-supervised benchmarks, we verify that MIRA improves over SwAV in various downstream tasks.

## 4 Experiments

In this section, we evaluate the representation quality learned via MIRA. We first provide the implementation details of our representation learning with MIRA (Sec. 4.1). We present our main results on linear, *k*-NN, semi-supervised learning, and transfer learning benchmarks in comparison to other self-supervised baselines (Sec. 4.2). Finally, we conduct an analysis of MIRA (Sec. 4.3).

### 4.1 Implementation details

We mostly follow the implementation details of representation learning from our baselines [8, 9]. More training details about evaluation procedures and analysis are described in the Appendix.

**Architecture**  The training model (network) consists of an encoder, projection head, and classification head as in Caron et al. [8]. We employ a widely used ResNet50 [29] as our base encoder and use the output of average-pooled 2048d embedding for representation training and downstream evaluations. The projection head is a 3-layer fully connected MLP of sizes $[2048, 2048, d]$; each hidden layer is followed by batch normalization [33] and ReLU. The classification head is used to predict the pseudo-labels; it is composed of an L2-normalization layer and a weight-normalization layer of the size $d \times K$. We use $d = 256$ and $K = 3000$.

**Training details**  We train our model on the training set of the ILSVRC-2012 ImageNet-1k dataset [18] without using class labels. We use the same data augmentation scheme (color jittering, Gaussian blur, and solarization) and multi-crop strategy (two $224 \times 224$ and six $96 \times 96$) used in Caron et al. [9]. We use a batch size of 4096 and employ the LARS optimizer [52] with a weight decay of $10^{-6}$. We use linearly scaled learning rate of $lr \times$ batch size$/256$ [27] with a base learning rate of $0.3$.[5] We adjust the learning rate with 10 epochs of a linear warmup followed by cosine scheduling. We also use an exponential moving average (EMA) network by default. When

---

[5]Otherwise stated, we also use a linearly scaled learning rate for evaluation training.

Table 3: *k*-**NN classification results on ImageNet with respect to subsets.** For 1% and 10% results, we evaluate the baselines by models of official codes. Baseline results are from [9].

| Method | ImageNet subset | | |
| | 100% | 10% | 1% |
|---|---|---|---|
| BYOL [28] | 64.8 | 57.4 | 45.2 |
| SwAV [8] | 65.7 | 57.4 | 44.3 |
| BarlowTwins [53] | 66.0 | 59.0 | 47.7 |
| DeepCluster-v2 [8] | 67.1 | 59.2 | 46.5 |
| DINO [9] | 67.5 | 59.3 | 47.2 |
| MIRA | **68.7** | **60.7** | **47.8** |

Table 4: **Semi-supervised learning results on ImageNet.** The baselines results are from [53]. Results style: **best**, second best

| | 1% | | 10% | |
| | Top-1 | Top-5 | Top-1 | Top-5 |
|---|---|---|---|---|
| Supervised | 25.4 | 48.4 | 56.4 | 80.4 |
| SimCLR [10] | 48.3 | 75.5 | 65.6 | 87.8 |
| BYOL [28] | 53.2 | 78.4 | 68.8 | 89 |
| SwAV [8] | 53.9 | 78.5 | **70.2** | 89.9 |
| BarlowTwins [53] | 55 | 79.2 | 69.7 | 89.3 |
| MIRA | **55.6** | **80.5** | 69.9 | **90.0** |

using EMA, we set the momentum update parameter to start from 0.99 and increase it to 1 by cosine scheduling. We use temperature scales of $\tau_s = 0.1, \tau_t = 0.225$ with a trade-off coefficient $\beta = 2/3$. We obtain soft pseudo-labels by 30 steps of the fixed point iteration. We further discuss this choice in Sec. 4.3. Otherwise stated, we use the encoder model trained for 400 epochs with multi-crop augmentations for the downstream task evaluations in this section.

## 4.2 Main results

**Linear evaluation** Tables 1 and 2 report linear evaluation results. We follow the linear evaluation settings in [28, 10]. We train a linear classifier on the top of the frozen trained backbone with the labeled training set of ImageNet. We train for 100 epochs using an SGD optimizer with a batch size of 1024. We choose a learning rate[6] with a local validation set in the ImageNet train dataset and adjust the learning rate by cosine annealing schedule. We apply random-resized-crop and horizontal flip augmentations for training. We evaluate the representation quality by the linear classifier's performance on the validation set of ImageNet.

Table 1 shows linear evaluation performance in top-1 accuracy for different training epochs. We train and report the performance of MIRA in two settings, with and without multi-crop augmentations. With multi-crop augmentations, MIRA consistently outperforms baselines while achieving 75.6% top-1 accuracy with only 400 epochs of training. We also report that 200 epochs of training with MIRA can outperform the 800 epochs results of other baselines that do not use multi-crops. Without multi-crop augmentations, MIRA performs slightly worse than BYOL [28]. However, MIRA performs the best among the clustering-based [6, 8] and information-driven [53, 26] methods.

In Table 2, we compare MIRA to other self-supervised methods with the final performance. MIRA achieves state-of-the-art performance on linear evaluation of ImageNet with only 400 epochs of training. While TWIST [26] can achieve similar performance to MIRA within 450 epochs, they require an extra training stage with *self-labeling*; without it, they achieve 74.1% accuracy with 800 epochs of training. In contrast, MIRA does not require additional training.

**Semi-supervised learning** In Table 4, we evaluate the trained model on the semi-supervised learning benchmark of ImageNet. Following the evaluation protocol in [28, 10], we add a linear classifier on top of the trained backbone and fine-tune the model with ImageNet 1% and 10% subsets. We report top-1 and top-5 accuracies on the validation set of ImageNet. For the 1% subset, MIRA outperforms the baselines; both the top-1 and top-5 accuracies achieve the best. For the 10% subset, MIRA is comparable to SwAV [8].

*k*-**NN evaluation** We evaluate the quality of learned representation via the nearest neighbor classifier. We follow the evaluation procedure of Caron et al. [9]. We first store the representations of labeled training data; then, we predict the test data by the majority vote of the *k*-nearest stored representations. We use 1/10/100% subsets of ImageNet training dataset to produce labeled representations. For ImageNet 1% and 10% subsets, we use the same subsets of semi-supervised learning evaluation. We

---

[6]We use the learning rate of 0.075 for multi-crop 400 epochs training.

Table 5: **Linear evaluation results on the transfer learning datasets.** Following Ericsson et al. [22], we report top-1 accuracy on Food, CIFAR-10/100, SUN397, Cars, DTD; mean-per-class accuracy on Aircraft, Pets, Caltech-101, Flowers; 11-point mAP metric on VOC2007. Results style: **best**

| | Aircraft | Caltech101 | Cars | CIFAR10 | CIFAR100 | DTD | Flowers | Food | Pets | SUN397 | VOC2007 | avg. |
|---|---|---|---|---|---|---|---|---|---|---|---|---|
| Supervised | 43.59 | 90.18 | 44.92 | 91.42 | 73.90 | 72.23 | 89.93 | 69.49 | **91.45** | 60.49 | 83.6 | 73.75 |
| InfoMin [46] | 38.58 | 87.84 | 41.04 | 91.49 | 73.43 | 74.73 | 87.18 | 69.53 | 86.24 | 61.00 | 83.24 | 72.21 |
| MoCo-v2 [13] | 41.79 | 87.92 | 39.31 | 92.28 | 74.90 | 73.88 | 90.07 | 68.95 | 83.3 | 60.32 | 82.69 | 72.31 |
| SimCLR-v2 [11] | 46.38 | 89.63 | 50.37 | 92.53 | 76.78 | 76.38 | 92.9 | 73.08 | 84.72 | 61.47 | 81.57 | 75.07 |
| BYOL [28] | 53.87 | 91.46 | 56.4 | 93.26 | 77.86 | 76.91 | 94.5 | 73.01 | 89.1 | 59.99 | 81.14 | 77.05 |
| DeepCluster-v2 [8] | 54.49 | 91.33 | 58.6 | 94.02 | **79.61** | **78.62** | 94.72 | 77.94 | 89.36 | 65.48 | 83.94 | 78.92 |
| SwAV [8] | 54.04 | 90.84 | 54.06 | 93.99 | 79.58 | 77.02 | 94.62 | 76.62 | 87.6 | 65.58 | 83.68 | 77.97 |
| MIRA | **59.06** | **92.21** | **61.05** | **94.20** | 79.51 | 77.66 | **96.07** | **78.76** | 89.95 | 65.84 | **84.10** | **79.86** |

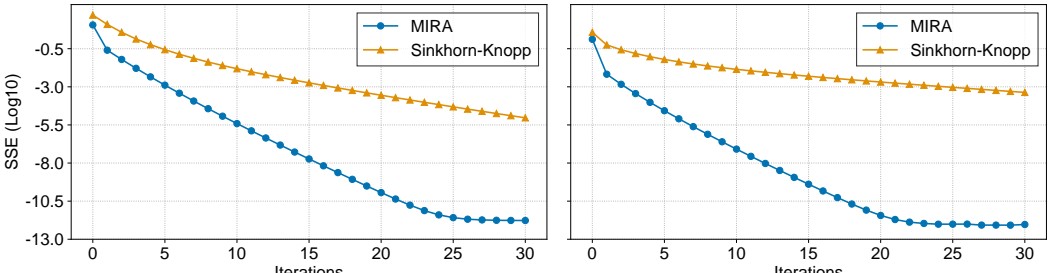

Figure 2: **Convergence analysis of MIRA and Sinkhorn-Knopp.** We observe the converging behavior of MIRA (**blue**) and Sinkhorn-Knopp (**yellow**). We experiment with trained models of MIRA (**left**) and SwAV (**right**). Since both methods are proven to converge, we iterate each method 1000 steps and regard the results as ground truth. We report the sum-squared error (SSE) with respect to the converging point in the log scale.

use the same hyperparameter settings in Caron et al. [9] with $k = 20$ nearest neighbors, temperature scaling [7] of $0.07$, and cosine distance metric.

Table 3 shows the classification accuracies on the validation set of ImageNet. The results show that our method achieves the best evaluation performance. Specifically, our method outperforms the previous state-of-the-art DINO [9] on 100% and 10% subset evaluation by 1.2 ∼1.4%. We note that BarlowTwins [53], a method also motivated by information maximization, shows a strong performance of 47.7% in the 1% subset evaluation.

**Transfer learning** We evaluate the representation learned by MIRA on the transfer learning benchmark with FGVC aircraft [41], Caltech-101 [25], Standford Cars [36], CIFAR-10/100 [37], DTD [15], Oxford 102 Flowers [42], Food-101 [5], Oxford-IIIT Pets [43], SUN397 [48], and Pascal VOC2007 [24] datasets. We follow the linear evaluation procedure from Ericsson et al. [22] that fits a multinomial logistic regression classifier on the extracted representations of 2048d from the trained encoder. We perform a hyperparameter search on the L2-normalization coefficient of the logistic regression model. The final performance is evaluated on the model that is retrained on all training and validation sets with the found coefficient.

Table 5 shows the performance of our algorithm compared to other baselines in 11 datasets. MIRA outperforms supervised representation on 10 out of 11 datasets. Compared to other self-supervised methods, representations learned from MIRA achieved the best performance in 9 out of 11 datasets, with an average improvement of 0.9% over the second-best baseline method. The results confirm that the representation trained with MIRA has a strong generalization ability for classification.

### 4.3 Analysis

**Convergence of pseudo-label assignment** We study the convergence speed of the proposed fixed point iteration in MIRA. We also experiment with the Sinkhorn-Knopp (SK) algorithm [16] used in SwAV [8] as a baseline. Both methods have experimented with a batch size of 512. We observe the

---

[7]The temperature scaling $\tau$ is used to calculate contributions $\alpha_i \sim \exp(\text{distance}_i/\tau)$ and voting is weighted by the contributions of the nearest neighbors.

converging behavior with the pre-trained models from MIRA and SwAV. Results are averaged over 1000 randomly sampled batches.

Figure 2 shows the result of the converging behavior of our method **(blue)** and SK algorithm **(yellow)** on trained models of MIRA **(left)** and SwAV **(right)**. Our fixed-point iteration converges faster than the SK algorithm in both pre-trained models, and our default setting of 30 steps of updates is sufficient for convergence. While we use 30 steps of updates to follow the theoretical motivation in our main experiments, we observe that choosing a small number of iterations is possible in practice.[8]

**Multi-crop and EMA**    Table 6 reports an ablation study on how EMA and multi-crop augmentations affect our representation quality. We train a model for 200 epochs in the settings with and without EMA or Multi-crop augmentations. EMA and Multi-crop augmentations greatly improve the linear evaluation performance as in Caron et al. [8, 9]. We take a further comparison with baselines that are in the same setups. While the only difference in the pseudo-labeling, our method outperforms SwAV [8] by 1.3% in top-1 accuracy. DINO [9] uses both multi-crop and EMA, our method outperforms DINO with fewer training epochs. The results validate the effectiveness of MIRA.

Table 6: **Ablation study about EMA and multi-crop augmentation.** We report top-1 accuracy with linear evaluation on validation set of ImageNet. The results of SwAV is from [30].

| Method | Multi-Crop | EMA | Epochs | Top-1 |
|--------|-----------|-----|--------|-------|
| SwAV | ✗ | ✗ | 200 | 69.1 |
| DINO | ✓ | ✓ | 300 | 74.5 |
| | ✗ | ✗ | 200 | 70.4 |
| MIRA | ✗ | ✓ | 200 | 72.3 |
| | ✓ | ✓ | 200 | 74.9 |

**Scalability**    We further validate MIRA's scalability on the small-and medium-scaled datasets. ResNet-18 is used as a base encoder throughout the experiments. While changing the base encoder, other architectural details remain the same as in ImageNet-1k. We do not apply multi-crop augmentations while using the EMA. We use image sizes of $32\times32$ and $256\times256$ for small and medium datasets, respectively. Following the procedures in da Costa et al. [17], we report the linear evaluation performance on the validation set. More experimental details about the optimizer, batch size, augmentations, etc., are provided in the Appendix.

Table 7: **Linear evaluation performance in small-and medium-scaled datasets.** We report top-1 and top-5 accuracies of linear evaluation on validation dataset. The training results are based on 1000 and 400 epochs of training on CIFAR-10/100 and ImageNet-100, respectively. Results style: **best**, second best

| Method | Arch. | CIFAR-10 | | CIFAR-100 | | ImageNet-100 | |
|--------|-------|----------|-------|-----------|-------|--------------|-------|
| | | Top-1 | Top-5 | Top-1 | Top-5 | Top-1 | Top-5 |
| BarlowTwins [53] | R18 | 92.10 | 99.73 | **70.90** | 91.91 | 80.16 | 95.14 |
| BYOL [28] | R18 | 92.58 | 99.79 | 70.46 | 91.96 | 80.32 | 94.94 |
| DeepCluster-v2 [8] | R18 | 88.85 | 99.58 | 63.61 | 88.09 | 75.36 | 93.10 |
| DINO [8] | R18 | 89.52 | 99.71 | 66.76 | 90.34 | 74.90 | 92.78 |
| SwAV [8] | R18 | 89.17 | 99.68 | 64.88 | 88.78 | 77.83 | 95.06 |
| MIRA | R18 | **93.02** | **99.87** | 70.65 | **92.23** | **81.00** | **95.56** |

The results are in Table 7. In CIFAR-10 and ImageNet-100, our method outperforms other self-supervised baselines by 0.4% and 0.7% in top-1 accuracy, respectively. For CIFAR-100, our method is comparable to the best performing baseline–BarlowTwins; MIRA performs better in top-5 accuracy.

**Training with small batch**    Throughout the experiments in Sec. 4.2, we use a batch size of 4096. While such batch size is commonly used in self-supervised learning, a large amount of GPU memory is required, limiting the accessibility. In Table 8, we test our method with a smaller batch size of

---

[8]The result with a small number of iterations is in the Appendix.

512 that can be used in an 8 GPU machine with 96GB memory. In this setting, we use the SGD optimizer with a weight decay of $10^{-4}$. We also test the robustness of pseudo-labeling with the Sinkhorn-Knopp algorithm in SwAV [8] and compare the results.

We report a top-1 linear evaluation performance of both methods after 100 epochs of training. In the result, the performance gap between our method and SwAV is amplified from 2.9% to 6% in the reduced batch size of 512. One possible explanation is that since SwAV is based on the equipartition constraint, the performance of SwAV harshly degrades when the batch size is not enough to match the number of clusters.

Table 8: **Linear evaluation performance with smaller batch size.** All results are based on ImageNet training. We also report the GPU memory usage and time spent for one epoch training. † is result by us.

| Method | Batch size | Epochs | GPU | GPU memory | Time per Epoch | Top-1 |
|---|---|---|---|---|---|---|
| SwAV† | 512 | 100 | $8 \times$ TITAN V | 71 GB | 23 min | 62.3 |
| MIRA w/o EMA | 512 | 100 | $8 \times$ TITAN V | 71 GB | 23 min | 66.3 |
| MIRA | 512 | 100 | $8 \times$ TITAN V | 73 GB | 29 min | 68.3 |
| SwAV [12] | 4096 | 100 | - | - | - | 66.5 |
| MIRA w/o EMA | 4096 | 100 | $16 \times$ A100 | 486 GB | 9 min | 68.7 |
| MIRA | 4096 | 100 | $16 \times$ A100 | 504 GB | 9 min | 69.4 |

## 5 Discussion

**Conclusion**   This paper proposes the mutual information maximization perspective pseudo-labeling algorithm MIRA. We formulate online pseudo-labeling into a convex optimization problem with mutual information regularization and solve it in a principled way. We apply MIRA for representation learning and demonstrate the effectiveness in various self-supervised learning benchmarks. We hope our simple yet theoretically guaranteed approach to mutual information maximization will guide many future applications.

**Limitation and negative social impact**   Our mutual information regularization with optimization strategy seems applicable to various tasks and domains, e.g., semi-supervised training [38]. However, we validate the effectiveness only in self-supervised visual representation learning. We note that the performance of MIRA for non-classification downstream tasks[9], e.g., detection, is not as dominating as in the classification tasks. In these tasks, methods that consider local or pixel-wise information achieve superior performance; incorporating such formulation into clustering-based methods seems to be an important future direction. Furthermore, despite our improved training efficiency, the self-supervised learning methods still require massive computation compared to supervised approaches. Such computational requirements may accelerate the environmental problems of global warming.

## Acknowledgments

This work was supported by Institute of Information & communications Technology Planning & Evaluation (IITP) grant funded by the Korea government(MSIT) (No.2022-0-00926). We thank Suyoung Lee, Minguk Jang, Dong Geun Shin, and anonymous reviewers for constructive feedbacks and discussions to improve our paper. Part of the experiments for this work was done at the Fundamental Research Lab of LG AI Research.

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
