# A  Appendix

## A.1  PyTorch pseudo-code for MIRA

---
**Algorithm 1** PyTorch pseudo-code of MIRA
---

```
# network(x): normalize(projector(encoder(x))) @ normalize(classifier.weights).T
# tau_t: temperature scale for target
# tau_s: temperature scale for training
# beta: MI-regularization coefficient

for x in loader:
    # Multi-view augmentations
    x1, x2 = aug(x), aug(x)
    logit1, logit2 = network(x1), network(x2)

    # pseudo-labeling -- MIRA
    target1 = mira(logit1, tau_t, beta)
    target2 = mira(logit2, tau_t, beta)

    # Swapped prediction loss
    loss = CrossEntropyLoss(logit1/tau_s, target2) + CrossEntropyLoss(logit2/tau_s, target1)
    loss.backward() # back propagation

    # Optimization
    update(network) # sgd updates

# The fixed-point iteration
def mira(logit, tau, beta, iters=30):
    k = softmax(logit / tau / (1 - beta), dim=1)
    v = k.mean(dim=0).pow(1 - beta)
    for _ in range(iters):
        temp = k/(v.pow(- beta) * k).sum(dim=1)
        v = temp.mean(dim=0).pow(1 - beta)
    v = v.pow(- beta / (1 - beta))
    target = v * k / (v * k).sum(dim=1)
    return target
```

---

## A.2  Proof of proposition ??

In this subsection, we derive the necessary and sufficient condition in proposition **??**. Denote $B, K$ be some natural numbers. We first prove the strict convexity of the optimization function $f : \mathbb{R}_+^{BK \times 1} \to \mathbb{R}$:

$$f(\boldsymbol{W}) = -\frac{1}{B} \sum_{i=1}^{B} \sum_{j=1}^{K} w_{ij} \log p_{ij} + \frac{1-\beta}{B} \sum_{i=1}^{B} \sum_{j=1}^{K} w_{ij} \log w_{ij} + \beta \sum_{j=1}^{k} \overline{w}_j \log \overline{w}_j, \qquad (1)$$

where $\overline{w}_j = \frac{1}{B} \sum_{i=1}^{B} w_{ij}$.

**Lemma 1.** *For $x \in \mathbb{R}_+^{N \times 1}$, $s(\boldsymbol{x}) = \sum_i^N x_i \log x_i$ is a strictly convex function of $\boldsymbol{x}$.*

*Proof.* Since the Hessian of $s$ is a diagonal matrix with positive elements $\nabla_{\boldsymbol{x}}^2 s(\boldsymbol{x})_{i,i} = 1/x_i$, $s$ is a strictly convex function. □

**Corollary 1.** *For $\boldsymbol{W} \in \mathbb{R}_+^{BK \times 1}$, $f(\boldsymbol{W})$ is a strictly convex function of $\boldsymbol{W}$.*

*Proof.* Note that

1. $f_1(\boldsymbol{W}) = -\frac{1}{B} \sum_{i=1}^{B} \sum_{j=1}^{K} w_{ij} \log p_{ij}$ is a affine transformation of $\boldsymbol{W}$,

2. $f_2(\boldsymbol{W}) = \frac{1-\beta}{B} \sum_{i=1}^{B} \sum_{j=1}^{K} w_{ij} \log w_{ij}$ is a strictly convex function of $\boldsymbol{W}$ by lemma 1,

3. $f_3(\boldsymbol{W}) = \beta \sum_{j=1}^{k} \overline{w}_j \log \overline{w}_j$ is a convex function of $\boldsymbol{W}$ since $f_3$ is the composition of the strictly convex function $s$ and affine transformation $\overline{w}_j = \frac{1}{B} \sum_{i=1}^{B} w_{ij}$.

The function $f(\boldsymbol{W}) = f_1(\boldsymbol{W}) + f_2(\boldsymbol{W}) + f_3(\boldsymbol{W})$ that is the sum of the convex and strictly convex terms becomes strictly convex. □

We show that the optimization function $f$ is strictly convex. The optimization problem (Eq. **??**) is defined on the convex set, i.e., $\boldsymbol{W} \subset \Delta_K := \{x \in \mathbb{R}_+^K \mid x^\mathsf{T} \mathbf{1}_K = 1\}$; hence, the problem is a *strictly*

*convex optimization problem*[1]. By the strict convexity, our optimization problem has a unique optimal point that has Karush–Kuhn–Tucker (KKT) conditions as the necessary and sufficient condition for optimality. By using the KKT condition, we derive our necessary and sufficient condition Eq. **??**.

Considering the constraint of the optimization domain, i.e., $\boldsymbol{W} \in \Delta_K \Leftrightarrow \forall i \in 1 : B, \sum_{j=1}^{K} w_{ij} = 1$, the Lagrangian function of our optimization problem becomes:

$$\boldsymbol{W}^* = \underset{\boldsymbol{W} \subset \Delta_K}{\arg\min} -\frac{1}{B} \sum_{i=1}^{B} \sum_{j=1}^{K} w_{ij} \log p_{ij} + \frac{1-\beta}{B} \sum_{i=1}^{B} \sum_{j=1}^{K} w_{ij} \log w_{ij} + \beta \sum_{j=1}^{k} \overline{w}_j \log \overline{w}_j \tag{2}$$

$$= -\underset{\boldsymbol{W} \subset \Delta_K}{\arg\min} \sum_{i=1}^{B} \sum_{j=1}^{K} \left[ w_{ij} \log p_{ij} + (1-\beta) w_{ij} \log w_{ij} + \beta w_{ij} \log \sum_{k=1}^{B} w_{kj} \right] \tag{3}$$

$$\Rightarrow L(\boldsymbol{W}, \boldsymbol{\lambda}) = \sum_{i=1}^{B} \sum_{j=1}^{K} \left[ w_{ij} \log p_{ij} + (1-\beta) w_{ij} \log w_{ij} + \beta w_{ij} \log \sum_{k=1}^{B} w_{kj} \right] + \sum_{i=1}^{B} \lambda_i (\sum_{j=1}^{K} w_{ij} - 1), \tag{4}$$

where $\boldsymbol{\lambda}$ is a Lagrange multipliers. By the KKT conditions with $\sum_{j=1}^{K} w_{ij} = 1$,

$$(\nabla_{\boldsymbol{W}} L(\boldsymbol{W}^*, \boldsymbol{\lambda}^*))_{ij} = -\log p_{ij} + (1-\beta)(1 + \log w_{ij}^*) + \beta(1 + \log \sum_{k=1}^{B} w_{kj}^*) + \lambda_i^* = 0 \tag{5}$$

$$\Leftrightarrow \log w_{ij}^* = \frac{1}{1-\beta} \left( \log p_{ij} - \lambda_i - 1 - \beta \log \sum_{k=1}^{B} w_{kj}^* \right) \tag{6}$$

$$\Leftrightarrow w_{ij}^* = p_{ij}^{\frac{1}{1-\beta}} (\sum_{k=1}^{B} w_{kj}^*)^{\frac{-\beta}{1-\beta}} \exp(\frac{-\lambda_i^* - 1}{1-\beta}) \tag{7}$$

$$\Leftrightarrow w_{ij}^* = \frac{p_{ij}^{\frac{1}{1-\beta}} (\sum_{k=1}^{B} w_{kj}^*)^{\frac{-\beta}{1-\beta}}}{\sum_{m=1}^{K} p_{im}^{\frac{1}{1-\beta}} (\sum_{k=1}^{B} w_{km}^*)^{\frac{-\beta}{1-\beta}}} = \frac{\overline{w^*}_j^{-\frac{\beta}{1-\beta}} p_{ij}^{\frac{1}{1-\beta}}}{\sum_{k=1}^{K} \overline{w^*}_k^{-\frac{\beta}{1-\beta}} p_{ik}^{\frac{1}{1-\beta}}}, \tag{8}$$

proves the necessary and sufficient condition in proposition **??**. The Eq. 8 comes from $\sum_j w_{ij}^* = 1$. For $\overline{w^*}$, this condition becomes:

$$\overline{w^*}_j = \frac{1}{B} \sum_{i=1}^{B} \omega_{ij}^* \tag{9}$$

$$= \frac{1}{B} \sum_{i=1}^{B} \frac{\overline{w^*}_j^{-\frac{\beta}{1-\beta}} p_{ij}^{\frac{1}{1-\beta}}}{\sum_{k=1}^{K} \overline{w^*}_k^{-\frac{\beta}{1-\beta}} p_{ik}^{\frac{1}{1-\beta}}} = \overline{w^*}_j^{-\frac{\beta}{1-\beta}} \frac{1}{B} \sum_{i=1}^{B} \frac{p_{ij}^{\frac{1}{1-\beta}}}{\sum_{k=1}^{K} \overline{w^*}_k^{-\frac{\beta}{1-\beta}} p_{ik}^{\frac{1}{1-\beta}}} \tag{10}$$

$$\Leftrightarrow \overline{w^*}_j = \left[ \frac{1}{B} \sum_{i=1}^{B} \frac{p_{ij}^{\frac{1}{1-\beta}}}{\sum_{k=1}^{K} \overline{w^*}_k^{-\frac{\beta}{1-\beta}} p_{ik}^{\frac{1}{1-\beta}}} \right]^{1-\beta}. \tag{11}$$

### A.3 Convergence of the fixed point iteration

In this subsection, we prove the convergence of our fixed point iteration (Eq. **??**). Denote $B, K$ be some natural numbers.

**Lemma 2.** *For $\boldsymbol{a}, \boldsymbol{x} \in \mathbb{R}_+^{B \times 1}$ and $\beta \in [0, 1)$, $h_{\boldsymbol{a}}(\boldsymbol{x}) := (\frac{1}{B} \sum_{i=1}^{B} a_i x_i^{-1})^{-\beta}$ is a concave function of $\boldsymbol{x}$.*

---

[1]In the main paper, we misrepresent our optimization problem as a strongly convex optimization problem; while it is a strictly convex optimization problem.

*Proof.* Denote $r_{\boldsymbol{a}}(\boldsymbol{x}) := \frac{1}{B}\sum_{i=1}^{B} a_i x_i^{-1}$ for a simplicity of expression; hence $h_{\boldsymbol{a}}(\boldsymbol{x}) = r_{\boldsymbol{a}}(\boldsymbol{x})^{-\beta}$. The second order partial derivatives of $h_{\boldsymbol{a}}(\boldsymbol{x})$ w.r.t. $\boldsymbol{x}$ becomes:

$$\Rightarrow (\nabla_{\boldsymbol{x}} h_{\boldsymbol{a}}(\boldsymbol{x}))_l = \beta r_{\boldsymbol{a}}(\boldsymbol{x})^{-\beta-1} \frac{a_l}{Bx_l^2} \tag{12}$$

$$\Rightarrow (\nabla_{\boldsymbol{x}}^2 h_{\boldsymbol{a}}(\boldsymbol{x}))_{l,m,l\neq m} = \beta(\beta+1) r_{\boldsymbol{a}}(\boldsymbol{x})^{-\beta-2} \frac{a_l}{Bx_l^2} \frac{a_m}{Bx_m^2} \tag{13}$$

$$(\nabla_{\boldsymbol{x}}^2 h_{\boldsymbol{a}}(\boldsymbol{x}))_{l,l} = \beta(\beta+1) r_{\boldsymbol{a}}(\boldsymbol{x})^{-\beta-2} \frac{a_l}{Bx_l^2} \frac{a_l}{Bx_l^2} - \beta r_{\boldsymbol{a}}(\boldsymbol{x})^{-\beta-1} \frac{2a_l}{Bx_l^3}. \tag{14}$$

For $\omega \in \mathbb{R}^{B\times 1}$,

$$\omega^{\mathsf{T}} \nabla_{\boldsymbol{x}}^2 h_{\boldsymbol{a}}(\boldsymbol{x})\omega = \beta(\beta+1) r_{\boldsymbol{a}}(\boldsymbol{x})^{-\beta-2} \sum_{l,m} \frac{a_l \omega_l}{Bx_l^2} \frac{a_m \omega_m}{Bx_m^2} - \beta r_{\boldsymbol{a}}(\boldsymbol{x})^{-\beta-1} \sum_l \frac{2a_l \omega_l^2}{Bx_l^3} \tag{15}$$

$$= \beta r_{\boldsymbol{a}}(\boldsymbol{x})^{-\beta-2} \left( (\beta+1)(\sum_i \frac{a_i \omega_i}{Bx_i^2})^2 - 2(\sum_i \frac{a_i}{Bx_i})(\sum_i \frac{a_i \omega_i^2}{Bx_i^3}) \right) \tag{16}$$

$$\leq \beta r_{\boldsymbol{a}}(\boldsymbol{x})^{-\beta-2} \left( (\beta+1)(\sum_i \frac{a_i |\omega_i|}{Bx_i^2})^2 - 2(\sum_i \frac{a_i}{Bx_i})(\sum_i \frac{a_i \omega_i^2}{Bx_i^3}) \right) \tag{17}$$

$$\leq -\beta(1-\beta) r_{\boldsymbol{a}}(\boldsymbol{x})^{-\beta-2} (\sum_i \frac{a_i |\omega_i|}{Bx_i^2})^2 \tag{18}$$

$$< 0. \tag{19}$$

The inequality in Eq. 18 comes from Cauchy–Schwartz inequality as bellows:

$$(\sum_i \frac{a_i}{Bx_i})(\sum_i \frac{a_i \omega_i^2}{Bx_i^3}) \geq (\sum_i \sqrt{\frac{a_i}{Bx_i} \frac{a_i \omega_i^2}{Bx_i^3}})^2 = (\sum_i \frac{a_i |\omega_i|}{Bx_i^2})^2. \tag{20}$$

Since $\forall \omega \in \mathbb{R}^{B\times 1}, \omega^{\mathsf{T}} \nabla_{\boldsymbol{x}}^2 h_{\boldsymbol{a}}(\boldsymbol{x})\omega < 0$, the Hessian of $h_{\boldsymbol{a}}(\boldsymbol{x})$ is negative definite, $h_{\boldsymbol{a}}(\boldsymbol{x})$ is concave. $\square$

**Corollary 2.** *For* $v \in \mathbb{R}_+^{K\times 1}, \boldsymbol{Q} \in \mathbb{R}_+^{B\times K}$, $g_j(\boldsymbol{v}; \boldsymbol{Q}) := h_{(\boldsymbol{Q}^{\mathsf{T}})_j}(\boldsymbol{Q}\boldsymbol{v}) = (\frac{1}{B}\sum_{i=1}^{B} \frac{q_{ij}}{\sum_{k=1}^{K} v_k q_{ik}})^{-\beta}$ *is a concave function of* $\boldsymbol{v}$.

*Proof.* Since $g_j$ is a composition of the concave function $h_{(\boldsymbol{Q}^{\mathsf{T}})_j}$ (by Lem. 2) and affine transformation with $\boldsymbol{Q}$; $g_j$ becomes concave. $\square$

We introduce the proposition from [8] that proves geometrical convergence of positive concave mapping.

**Proposition 1** (Piotrowski and Cavalcante [8], Proposition 3). *Let* $f : \mathbb{R}_+^M \to int(\mathbb{R}_+^M)$ *be a continuous and concave mapping w.r.t cone order with a fixed point* $x^* \in int(\mathbb{R}_+^M)$. *Then, the fixed point iteration of* $f$, *i.e.,* $x_{n+1} = f(x_n)$, *with* $x \in \mathbb{R}_+^M$ *converges geometrically to* $x^*$ *with a factor* $c \in [0, 1)$.

By the proposition 1 and corollary 2, we prove the convergence of the following fixed point iteration:

**Proposition 2.** *For* $\beta \in [0, 1)$, $\boldsymbol{Q} \in \mathbb{R}_+^{B\times K}$, *and for any initialization* $v^{(0)} \in \mathbb{R}_+^{K\times 1}$, *the fixed point iteration:*

$$v_j^{(n+1)} = g_j(\boldsymbol{v}^{(n)}; \boldsymbol{Q}) = \left[ \frac{1}{B} \sum_{i=1}^{B} \frac{q_{ij}}{\sum_{k=1}^{K} v_k^{(n)} q_{ik}} \right]^{-\beta}, \tag{21}$$

*converges to the fixed point.*

*Proof.* We can see that output of the fixed point iteration $g_j(\boldsymbol{v}^{(n)}; \boldsymbol{Q})$ is always positive. By corollary 2, $g(\boldsymbol{v}^{(n)}; \boldsymbol{Q})$ is a concave mapping. Therefore, by proposition 1, the fixed point iteration converges to the fixed point. $\square$

Finally, by bijective change of variables $v_j^{(n)} = \left(u_j^{(n)}\right)^{-\beta/(1-\beta)}$,

$$v_j^{(n+1)} = g_j(\boldsymbol{v}^{(n)}; \boldsymbol{Q}) = \left[\frac{1}{B}\sum_{i=1}^{B}\frac{q_{ij}}{\sum_{k=1}^{K}v_k^{(n)}q_{ik}}\right]^{-\beta} \tag{22}$$

$$\Leftrightarrow \left(u_j^{(n+1)}\right)^{-\beta/(1-\beta)} = \left[\frac{1}{B}\sum_{i=1}^{B}\frac{q_{ij}}{\sum_{k=1}^{K}(u_k^{(n)})^{-\beta/(1-\beta)}q_{ik}}\right]^{-\beta} \tag{23}$$

$$\Leftrightarrow u_j^{(n+1)} = \left[\frac{1}{B}\sum_{i=1}^{B}\frac{q_{ij}}{\sum_{k=1}^{K}(u_k^{(n)})^{-\beta/(1-\beta)}q_{ik}}\right]^{1-\beta}, \tag{24}$$

we derive our fixed point iteration in Eq. **??**. Throughout updates, Eq. 24 continuously satisfies the relation $v_j^{(n)} = \left(u_j^{(n)}\right)^{-\beta/(1-\beta)}$ to the fixed point iteration by Eq. 21; hence, the fixed point iteration by Eq. 24 also converges to the fixed point. Furthermore, the fixed point by Eq. 24 satisfies the necessary and sufficient condition (Eq.11); is equal to $\overline{w^*}$.

## A.4 Implementation & evaluation details

For the transfer learning evaluation, we use the open-source implementation of Ericsson et al. [6] at `https://github.com/linusericsson/ssl-transfer`. For the k-NN evaluation, we refer to the official implementation of Caron et al. [2] at `https://github.com/facebookresearch/dino`. For the CIFAR-10/100 and ImageNet-100 training & evaluations, we refer to the implementations of da Costa et al. [5] at `https://github.com/vturrisi/solo-learn`.

### A.4.1 ImageNet-1k

**Representation learning**   We describe more detailed implementation details here. When we pre-train with EMA, we use the EMA backbone for downstream evaluations. We exclude BN parameters and biases from weight decay in the LARS optimizer. For 100/200 epochs training, we reduce the warm-up epochs to 5. Similar to temperature annealing of Caron et al. [2], we apply cosine annealing on $\beta$ from 0.7 to 2/3. For hyper-parameters tuning, we use ImageNet-100 tuned hyper-parameters ($\tau_t = 0.225, \beta = 2/3$) accross all ImageNet-1k, ImageNet-100, and CIFAR-10/100 datasets. For other settings, including the projector design, number of clusters, and classification (prototype) head, we follow SwAV implementation [1, 3].

**Semi-supervised learning**   We explain our semi-supervised learning evaluation training. We train 20 epochs with a batch size of 256. We employ an SGD optimizer with a cosine learning rate schedule. For 1% training, we find that freezing the backbone encoder (with 0. learning rate) performs the best; we use learning rates of 0.4 for the linear classifier. For 10% training, we use learning rates of 0.02 and 0.1 for the backbone encoder and linear classifier.

**More details about the linear evaluation**   As the local validation set (in the training dataset), we use the 1% training split from SimCLR in the ImageNet train set as our local validation set. We do not use regularization methods such as weight decay, gradient clipping, etc.

While we follow Grill et al. [7] for linear evaluation in the main results (Sec.**??**), it requires much computation. Thus, for the additional linear evaluations in the analysis section (Sec. **??**) and Appendix, we fix to use a LARS optimizer with a learning rate of 0.1 since it performs well regardless of different models and matches the performance with the SGD optimizer. For reference, we report MIRA's linear evaluation performance with the LARS optimizer.

Table 1: **The linear evaluation results with a LARS optimizer**.

| Method | Epochs | | | |
| --- | --- | --- | --- | --- |
| | 100 | 200 | 400 | 800 |
| MIRA (without multi-crop) | 69.4 | 72.1 | 72.9 | 73.8 |
| MIRA (with multi-crop) | 73.5 | 74.8 | 75.5 | - |

### A.4.2 CIFAR-10/100 and ImageNet-100

**Representation learning**   For experiments on CIFAR-10/100 and ImageNet-100, we follow the settings in da Costa et al. [5]. We use the same settings of ImageNet-1k experiments except for the base encoder, optimizer, augmentation scheme, and batch size. We employ ResNet18 as a base encoder for experiments on CIFAR-10/100 and ImageNet-100. For CIFAR-10/100 datasets, we change the first $7 \times 7$ convolution layer with stride $2$ into $3 \times 3$ convolution layer with stride $1$. We use the SGD optimizer with a weight decay of $10^{-4}$. We employ a linearly scaled learning rate with a base learning rate of $0.3$ as in ImageNet-1k and scheduled the learning rate with 10 epochs of a linear warmup followed by cosine scheduling. For CIFAR-10/100 datasets, we remove the GaussianBlur and adjust the minimum scale of RandomResizedCrop to $0.2$. We use batch sizes of 256 and 512 for CIFAR10/100 and ImageNet-100, respectively. We train for $1000$ and $400$ epochs for CIFAR-10/100 and ImageNet-100, respectively.

**Linear evaluation**   Following the da Costa et al. [5], we report online and offline linear evaluation results for CIFAR-10/100 and ImageNet-100, respectively. For online linear evaluation on CIFAR-10/100, we use the SGD optimizer with a learning rate of $0.1$. We do not apply weight decay and use cosine scheduled the learning rate. For the linear evaluation on ImageNet-100, we use the LARS optimizer with learning rate of $0.1$ and batch size of $1024$.

## A.5   Standard deviations

We report the standard deviations for our main results. The linear, semi-supervised, and transfer learning evaluations are conducted four times with four different random seeds.

Table 2: **The results on the linear and semi-supervised evaluations with standard deviations.**

| Linear | | Semi 1% | | Semi 10% | |
|---|---|---|---|---|---|
| Top-1 | Top-5 | Top-1 | Top-5 | Top-1 | Top-5 |
| 75.61 (0.036) | 92.50 (0.02) | 55.62 (0.032) | 80.46 (0.041) | 70.11 (0.055) | 89.9 (0.046) |

Table 3: **The results on the transfer learning evaluation with standard deviations.**

| Aircraft | Caltech101 | Cars | CIFAR10 | CIFAR100 | DTD | Flowers | Food | Pets | SUN397 | VOC2007 |
|---|---|---|---|---|---|---|---|---|---|---|
| 58.93 (0.16) | 92.11 (0.035) | 60.64 (0.28) | 94.20 (0.021) | 79.61 (0.071) | 77.62 (0.025) | 96.16 (0.037) | 78.84 (0.087) | 89.95 (0.23) | 65.84 (0.0019) | 84.10 (0.) |

## A.6   Results on 800 epochs training

We report MIRA 800 epochs training results on the linear and $k$-NN evaluations. To account for longer epoch training, we use the initial EMA momentum value of 0.996 as in Chen et al. [4]. For the linear evaluation of 800 epochs training, we follow the evaluation protocols in the main results [7].

Table 4: **MIRA 800 epochs training results on the linear and $k$-NN evaluations.**

| Epochs | Linear | k-NN (100%) | k-NN (10%) | k-NN (1%) |
|---|---|---|---|---|
| 400 | 75.6 | 68.7 | 60.7 | 47.8 |
| 800 | 75.7 | 68.8 | 61.1 | 48.2 |

## A.7   Experiments on the detection and segmentation task

We test our method on detectionsegmentation of the COCO 2017 dataset with Masked R-CNN, R50-C4 on a 2x scheduled setting. We use the configuration from the MoCo official implementation. MIRA performs better than the supervised baseline and is comparable to MoCo; it is not as dominating as in the classification tasks.

Table 5: **Detection and segmentation results on the COCO 2017 dataset.**

|      | $AP^{bb}$ | $AP^{bb}_{50}$ | $AP^{bb}_{75}$ | $AP^{mk}$ | $AP^{mk}_{50}$ | $AP^{mk}_{75}$ |
|------|-----------|----------------|----------------|-----------|----------------|----------------|
| Sup  | 40        | 59.9           | 43.1           | 34.7      | 56.5           | 36.9           |
| MoCo | 40.7      | 60.5           | 44.1           | 35.4      | 57.3           | 37.6           |
| MIRA | 40.6      | 61             | 44.1           | 35.3      | 57.2           | 37.3           |

## A.8 Ablation studies

### A.8.1 Ablation study on the number of clusters

In table 6, we study the effect of the number of clusters on the performance of linear and $k$-NN evaluation (with 1%/10%/100% labels). We train MIRA on ImageNet-1k for 100 epochs without multi-crop augmentations while varying the number of clusters. When the number of clusters is sufficiently large ($\geq 3000$), we observe no particular gain by varying the number of clusters. This is consistent with the observation in SwAV [1].

Table 6: **The linear$k$-NN evaluation results while varying the number of clusters**.

| # of clusters | 300  | 1000 | 3000 | 10000 | 30000 |
|---------------|------|------|------|-------|-------|
| Linear        | 67.7 | 69   | 69.3 | 69.5  | 69.5  |
| $k$-NN (100%) | 58.9 | 60.5 | 61.6 | 61.7  | 61.7  |
| $k$-NN (10%)  | 49.5 | 51.9 | 53.3 | 53.3  | 53.3  |
| $k$-NN (1%)   | 36.6 | 39.7 | 41.0 | 41.0  | 41.0  |

### A.8.2 Ablation study on the number of the fixed point iteration

We report the 100 and 400 epochs pre-training linear evaluation results with 1 and 3 fixed point iterations. In the 100 epochs training, the models with a smaller number of fixed point iterations, 1it and 3it, perform slightly better (+0.2%); in 400 epochs training, the model with more iterations, 30it, performs better (+0.2~0.3%). The convergence to the fixed point is not the primary factor for learning; however, choosing a sufficiently large number of fixed point iterations in longer epoch training seems reasonable.

Table 7: **The linear evaluation results of MIRA while varying the fixed point iteration steps**.

| # of iterations | 1it  | 3it  | 30it |
|-----------------|------|------|------|
| 100 epochs      | 69.5 | 69.5 | 69.3 |
| 400 epochs      | 72.7 | 72.6 | 72.9 |