# OpenReview forum: "Unsupervised Visual Representation Learning via Mutual Information Regularized Assignment"
_NeurIPS.cc/2022/Conference — NeurIPS 2022 Accept_

### Official Review · Reviewer_y6Mg · 2022-07-06

**Rating:** 6
**Confidence:** 4
**Soundness:** 3 good
**Presentation:** 2 fair
**Contribution:** 3 good

**Summary:**

The paper proposes a method of pseudo-labeling such that the pseudo-labels have maximal mutual information with the samples. This is in contrast to SWaV where the pseudo-labels have maximum entropy. The paper proposes an iterative algorithm to recover the pseudo-labels, which is used inside a SSL method, as done in SWaV.

**Questions:**

* What are the MIRA numbers when using the online network for probing the representation for all epochs?
* What are the test performances of MIRA, without re-training the linear probe, on ImageNet-v2? The author can compare with [2]
* What is the performance of MIRA with only 3 iterations?
* What are the practical reasons we should care about the convergence of the pseudo-labeling?
* In Table 6, are the hyper-parameters of SwAV and MIRA *exactly* the same except for the pseudo-labeling? If not, what are the results if all of the hyper-parameters are exactly the same (including the number of iteration of the pseudo-labeling algorithm).

**Limitations:**

The authors have discussed generic limitations of SSL methods, but they have not discussed limitations related to their method.

**Strengths And Weaknesses:**

The method proposes an interesting perspective on pseudo-labeling that was not previously discussed in the literature, as far as I know. Their methods is based on the KKT optimality conditions and the pseudo-labels are found
by iterating such condition. I appreciated the connection, but I believe that this connection is important and should be presented in the main paper. There is a clear connection with SWaV and I believe that they can be connected through the framework of Optimal Transport. I would have appreciated a more in depth discussion comparing
MIRA (the proposed method) with SWaV other than just saying that MIRA also minimizes the conditional entropy (while SWaV only maximizes the entropy). For example, could it be possible to define the updates of MIRA as a Sinkhorn algorithm
as it is done in the lecture of [0]. Other than this, I don't have much issues with the method section other than I believe that the authors should explain how they obtain their necessary and sufficient condition of Equation 4 in the main paper.

My biggest issues with this paper are with the experiments. The authors claim state-of-the art for their method for several setups. But, their are several issues with this claim:
* The authors select different hyper-parameters for the different training setups. E.g. they use different learning rates for 100/200 epochs in comparison to 400 and 800 epochs. This is not done for most of their baseline. E.g. BYOL [1] results are taken from the SimSiam paper and they used the same hyper-parameters for every setups.
* The results are reported on the momentum encoder, while the results of the other methods are generally reported on the online network. The results should be reported on the same network or it should be clear where the results are probed for all the methods.
* Looking at the code, it is not clear that the authors have not tuned their method on the validation/test set. This is not standard practice, see e.g. BYOL [1]
* The above is true especially for the parameters chosen for semi-supervised learning for ImageNet-1k.
* It is not clear what and how the linear classification hyper-parameters were chosen for ImageNet-1k.

The above concerns make it hard to be satisfied that the reported improvements are solely due to the proposed method, MIRA. Perhaps, if the authors made it clear that they used exactly the same parameters than SwAV and the the only intervention is the MIRA algorithm replacing the Sinkhorn, then it would be more satisfying.

Another concern that I have regards Figure 2. SwAV uses only 3 Sinkhorn iterations.
However, from Figure 2, it seems that, at that point, SwAV is far from converging and the authors off SwAV claim that this amount of iterations is sufficient for obtaining their better numbers. Then, is convergence to the fixed point really important for downstream performance?
If so, it is not clear to me why we should care about such speed of convergence. And, I am also wondering how MIRA is comparing to SWaV with a much smaller number of iteration. Is 30 iterations necessary?

Finally, I found the abstract and the introduction lacking clarity. For example, the phrase "We derive a fixed-point iteration method and prove its convergence to the optimal solution." does not tell much
about the algorithm. I believe that the authors should be more explicit about the algorithm and the idea presented in the paper in the introduction so that the reader have a general idea of the paper after reading the introduction.
In particular, I believe that the fourth paragraph of the introduction should be clearer about the method.

Minor: The learning rate is missing at L:427.

[0] https://bicmr.pku.edu.cn/~wenzw/bigdata/lect-ot.pdf
[1] https://arxiv.org/pdf/2006.07733.pdf

---

> ### Author Response · Authors · 2022-08-02
> **Response to reviewer y6Mg - 1**
>
> We appreciate the reviewer for the extensive feedback and the acknowledgment of the novelty of our proposed algorithm, MIRA. We hope our response will address the concerns.
>
> ---
>
> We firstly present our detailed response to the issues raised about our experiments. We will continuously clarify and address these issues in the revision.
>
> > I1. The authors select different hyper-parameters for the different training setups, e.g., they use different learning rates for 100/200 epochs in comparison to 400 and 800 epochs.
>
> &rarr; We respectfully clarify that we do not use different learning rates for 100/200 epochs in comparison to 400/800 epochs. We fix the base learning rate of 0.3 across all epochs; while we only reduce the warm-up epochs from 10 to 5 for 100/200 epochs training. To address the reviewer's concern, we perform an ablation study by setting MIRA on 100/200 epochs of training with 10 epochs of warm-up. The linear evaluation performances with 10 epochs warm-up match the reported performance in the main paper.
>
> |  Warm-up  | 100ep | 200ep |
> |:---------:|:-----:|:-----:|
> |  5 epochs |  69.4 |  72.1 |
> | 10 epochs |  69.3 |  72.1 |
>
> > I2 (Q1). MIRA performance with the online encoder for all epochs?
>
> &rarr; We report the linear evaluation performance with online encoders from pre-trained models for all epochs. The table below summarizes the result. While the usage of the representation from momentum encoders shows improvement over the usage of online encoders, its gain is limited to at most 0.2%. We believe this is not a dominant factor in MIRA's improvement. Our method still achieves 75.4% accuracy with only 400 epochs of training while using the online encoder.
>
> |  Method / Epochs |  100 |  200 |  400 |  800 |
> |:----------------:|:----:|:----:|:----:|:----:|
> |       MIRA       | 69.4 | 72.1 | 72.9 | 73.8 |
> |    MIRA-online   | 69.4 |  72  | 72.9 | 73.6 |
> |     MIRA (multi-crop)    | 73.5 | 74.8 | 75.5 |      |
> | MIRA-online (multi-crop) | 73.2 | 74.6 | 75.4 |      |
>
> > I3. It is not clear that the authors have not tuned their method on the validation/test set.
>
> &rarr; We do not tune our method on the validation/test sets. For semi-supervised evaluation, we conduct a hyper-parameter search over learning rates on a partial training set while following the other settings to SwAV. For linear evaluation, we do report the performance on various epochs and configurations; hence, we do not conduct any hyper-parameter search but adapt the optimizer (LARS) and base learning rate (0.1) settings from the official SimSiam implementation. Therefore, we speculate that further hyper-parameter search will strengthen our result on linear evaluation.

---

> > ### Author Response · Authors · 2022-08-02
> > **Response to reviewer y6Mg - 2**
> >
> > ---
> >
> > We secondly respond to the questions as follows:
> >
> > > Q1. please see the response in I2
> >
> > > Q2. What are the test performances of MIRA, without re-training the linear probe, on ImageNet-v2? The author can compare with [2]
> >
> > &rarr; The test performance of our method, without re-training the linear classification head, on ImageNet-v2 is as follows:
> > |  | ImageNet val | MatchedFrequency | Threshold0.7 | TopImages |
> > |:---:|:---:|:---:|:---:|:---:|
> > | MIRA | 75.5 | 62.9 | 71.6 | 76.9 |
> >
> > We note to the reviewer that the reference [2] is omitted in the review. We will make a comparison and do an analysis as soon as the reference [2] is provided.
> >
> > > Q3,4. What is the performance of MIRA with only 3 iterations? What are the practical reasons we should care about the convergence of the pseudo-labeling?
> >
> > &rarr; We report the 100 epochs pre-training linear evaluation results with 1 and 3 fixed point iterations.
> >
> > | # of iterations | 1it | 3it | 30it |
> > |:---:|:---:|:---:|:---:|
> > | MIRA 100ep | 69.5 | 69.5 | 69.3 |
> >
> > In the table, the convergence to the fixed point doesn't seem really important for downstream performance; hence, it seems possible to choose a small number of iterations in practice. However, in this case, it is ambiguous what we are using as a pseudo-label; in the experiment, we want our method to follow the theoretical motivation. ~~Furthermore, it is uncertain if such results hold when loss becomes sufficiently small as it approaches convergence~~. ~~We will later update the results on a longer epoch (when close to convergence) during the discussion period~~. We update the results with the longer epochs (400) training in "*Response to reviewer y6Mg - 2.1*".
> >
> > > Q5. In Table 6, are the hyper-parameters of SwAV and MIRA exactly the same except for the pseudo-labeling? If not, what are the results if all of the hyper-parameters are exactly the same.
> >
> > &rarr; They share most of the hyper-parameters, e.g., # of clusters, projection head, optimizer, weight decay, etc., but they are not exactly the same. Especially, MIRA and SwAV use different augmentation recipes; We follow the augmentation scheme of DINO cause it is more widely adopted. We report the SwAV's linear evaluation performance when all of the hyper-parameters are exactly the same including the augmentation recipe except for pseudo-labeling.
> >
> > | | MIRA (wo EMA) | SwAV | SwAV (SimSiam) |
> > |:---:|:---:|:---:|:---:|
> > | 100ep, Top-1 | 68.7 | 66.9 | 66.5 |
> >
> > While performance is increased, our method still performs better with a +1.8 % margin.

---

> > > ### Author Response · Authors · 2022-08-02
> > > **Response to reviewer y6Mg - 3**
> > >
> > > ---
> > > Thirdly, we clarify the comments as follows.
> > >
> > > > C1. The connection of “KKT conditions and iterative method to get the feasible solution” is important and should be presented in the main paper.
> > >
> > > &rarr; Indeed, we skip the derivation from the KKT condition (Eq. 4) to the iterative method (Eq. 5) in the main paper. We will incorporate the derivation below into the main paper.
> > >
> > > *By substituting the necessary and sufficient condition (Eq. 4) into* $\overline{w}\_j=\frac{1}{B}\sum\_{i=1}^B w\_{ij}$, *we get the necessary and sufficient condition with* $\overline{w^*}$.
> > >
> > > $$ \overline{w^\*}\_j = \frac{1}{B}\sum\_{i=1}^B w^\*_{ij}
> > >     = \frac{1}{B}\sum\_{i=1}^B \frac{\overline{w^\*}\_j^{-\frac{\beta}{1-\beta}} p\_{ij}^{\frac{1}{1-\beta}}}{\sum\_{k=1}^K \overline{w^\*}\_k^{-\frac{\beta}{1-\beta}} p\_{ik}^{\frac{1}{1-\beta}}} = \overline{w^\*}\_j^{-\frac{\beta}{1-\beta}} \frac{1}{B}\sum\_{i=1}^B  \frac{ p\_{ij}^{\frac{1}{1-\beta}}}{\sum\_{k=1}^K \overline{w^\*}\_k^{-\frac{\beta}{1-\beta}} p\_{ik}^{\frac{1}{1-\beta}}} \Leftrightarrow \overline{w^\*}\_j = \Bigg[ \frac{1}{B}\sum\_{i=1}^B  \frac{ p\_{ij}^{\frac{1}{1-\beta}}}{\sum\_{k=1}^K \overline{w^\*}\_k^{-\frac{\beta}{1-\beta}} p\_{ik}^{\frac{1}{1-\beta}}} \Bigg]^{1 - \beta}. $$
> > >
> > > *And the RHS of the last equation becomes the update rule (Eq. 5).*
> > >
> > > > C2. Authors should explain how they obtain their necessary and sufficient condition of Equation 4 in the main paper.
> > >
> > > &rarr; We will add an explanation--"*The proposition is driven by proving the strict convexity and then applying the KKT condition.*" around line 132.
> > >
> > > > C3. The abstract and the introduction lack clarity.
> > >
> > > &rarr; We will add further explanations on the flow of the ideas in the main paper and clarify our method in the introduction and abstract in the revised version.
> > >
> > > Especially for line 45, we will change into "*We formulate the problem as a strictly convex optimization problem and derive the necessary and sufficient condition of solution with the Karush-Kuhn-Tucker (KKT) condition. The solution can be achieved by fixed-point iteration.*".
> > >
> > > > C4. About "*a more in depth discussion comparing MIRA with SwAV other than just saying that MIRA also minimizes the conditional entropy (while SwAV only maximizes the entropy)*".
> > >
> > > &rarr; We further clarify the connection and difference of MIRA against the popular Sinkhorn algorithm. In fact, as the reviewer pointed out, the derivation of MIRA shares similar practices with Sinkhorn's algorithm. Both methods formulate the necessary and sufficient condition of the convex optimization problem via the KKT condition; use the condition to design an iterative method that converges to a feasible solution. However, differently from our problem (Eq. 3), the optimal transport problems are defined on the set of couplings; hence, it seems that our problem is not directly solvable via Sinkhorn's algorithm unless we find a way to reformulate the problem on the set of couplings. We agree that it will be an interesting direction to study.
> > >
> > > > Minor: The learning rate is missing at L:427.
> > >
> > > &rarr; In line 427, we set the learning rate into 0. We again clarify that this is just 0, not the typo.

---

> > > ### Author Response · Authors · 2022-08-07
> > > **Response to reviewer y6Mg - 2.1**
> > >
> > > (Additional response about Q3,4.)
> > > > Q3,4. What is the performance of MIRA with only 3 iterations? What are the practical reasons we should care about the convergence of the pseudo-labeling?
> > >
> > > We further experiment if the number of fixed point iterations matters when the model is trained with longer epochs (400). The results of the linear evaluation performance are as below:
> > >
> > > | # of iterations | 1it | 3it | 30it |
> > > |:---:|:---:|:---:|:---:|
> > > | MIRA, 100ep | 69.5 | 69.5 | 69.3 |
> > > | MIRA, 400ep | 72.7 | 72.6 | 72.9 |
> > >
> > > In the 100 epochs training, the models with a smaller number of fixed point iterations, 1it and 3it, perform slightly better (+0.2%); while in 400 epochs training, the model with more iterations, 30it, performs better (+0.2~0.3%). As a result, the convergence to the fixed point is not the major factor for learning; however, it seems reasonable to choose a sufficiently large number of fixed point iterations in longer epoch training.
> > >
> > > We further respond to the “*Why should we care about the speed of convergence*?“. In the experiments, we want our computed pseudo-labels to be close enough to the theoretically motivated ones in the method (section 3). By Prop. 3 in the Appendix, we prove the convergence of the fixed point iteration (Eq. 5); while the converging speed is unknown. Hence, we verify in Fig. 2 that our fixed point iteration converges fastly enough to use in practice to solve the optimization problem (Eq. 3). Besides that, we want to prove that our fixed point iteration is at least more effective than Sinkhorn's method, in which the SwAV is based on. Furthermore, our experimental results suggest that the more converged (with a larger number of iterations) gets better results on longer epochs training.

---

> > ### Author Response · Authors · 2022-08-08
> > **Response to reviwer y6Mg - 1.1**
> >
> > (Additional response about I3)
> >
> > The reviewer questioned the tuning of MIRA. To address more on this, we further follow BYOL's linear evaluation protocol and report the results in detail. Based on the practices of BYOL, we use SGD with a Nesterov momentum of 0.9 using a batch size of 1024 and sweep over 5 learning rates {0.1, 0.2, 0.3, 0.4, 0.5} (without linear scaling) on a local validation set in the ImageNet train set. Since the authors of BYOL do not specify the local validation set, we simply use the 1% training split from SimCLR in the ImageNet train set as our local validation set. We do not use regularization methods such as weight decay, gradient clipping, etc.; we adapt widely used the 100 epochs training and cosine scheduling to adjust learning rates as in SimSiam, SwAV, DINO, BarlowTwins, etc. The table below shows the linear evaluation result with 400 epochs trained model on the local validation set w.r.t. 5 learning rates.
> >
> > | lr | 0.1 | 0.2 | 0.3 (Best) | 0.4 | 0.5 |
> > |:---:|:---:|:---:|:---:|:---:|:---:|
> > | Top-1 Acc | 78.08 | 78.54 | 78.86 | 78.71 | 78.57 |
> >
> > Finally, we check the performance of the chosen hyperparameter (lr=0.3) by evaluating it on the test dataset. In the below table, we show the original result from the paper denoted as LARS, and the result by the chosen learning rate denoted as SGD. In both Top-1 and-5 accuracies, SGD shows a slight improvement of +0.1%.
> >
> > |  | Top-1 Acc | Top-5 Acc |
> > |---|:---:|:---:|
> > | SGD (new) | 75.6 | 92.6 |
> > | LARS (Paper) | 75.5 | 92.5 |
> >
> > We will contain the results in the paper and clarify the evaluation protocol in more detail.

---

### Official Review · Reviewer_E9n2 · 2022-07-08

**Rating:** 8
**Confidence:** 4
**Soundness:** 4 excellent
**Presentation:** 4 excellent
**Contribution:** 4 excellent

**Summary:**

This paper proposes a new self-supervised method for representation learning from unlabeled images. The core representation learning algorithm, called MIRA, relies on predicting pseudo labels from the data, which are chosen, actually learned, based on maximizing the mutual information between themselves and the data distribution. To make this proposed algorithm tractable, they derive a fixed-point iteration solution, for which they provide a proof for its ability to find the optimal solution. To achieve this, they prove that the proposed optimization problem for finding the optimal set of pseudo labels is a convex problem.
The proposed method is evaluated by assessing the quality of the learned data representations on multiple downstream tasks, obtained from several datasets. The experimental results highlight the richness of the learned semantic representations.

**Questions:**

Please see weaknesses above.

**Limitations:**

The authors provide these limitations adequately.

**Strengths And Weaknesses:**

Strengths:
1- The proposed method is novel, elegantly formulated, and encourages learning rich semantic representations with shorter training schedules compared to the baselines.
2- The writing is clear, and the used language is understandable.
3- The experimental evaluation is extensive, and is performed on several benchmarks and datasets. The results illustrate the merits of the method in improving the quality of the learned representations.
4- Essential to me is that this work also provides experimental results using smaller batch sizes, showed in Table 8. This set of results, while it can be a little more comprehensive, is necessary, from my point of view. It encourages a wider adoption of self-supervised learning for budget resources.

Weaknesses:
- I believe the number of clusters K, which is the number of pseudo label assignments, is used as a fixed number (3000) across the experimental results section. It would be interesting to study the effect of this hyper-parameter on the performance on downstream tasks.

---

> ### Author Response · Authors · 2022-08-02
> **Response to reviewer E9n2**
>
> We are glad that you not only acknowledged our methodology and experiments but also praised our additional experiments with small batch sizes. We address your question as follows:
>
>
> > Q1. It would be interesting to study the effect of this hyper-parameter on the performance on downstream tasks.
> >
> &rarr; Following the reviewer's comment, we study the effect of the number of clusters on the performance of linear and k-NN evaluation (with 1%/10%/100% labels). We train MIRA on ImageNet-1k for 100 epochs without multi-crop augmentations while varying the number (#) of clusters. When the number of clusters is sufficiently large enough (>=3000), we observe no particular gain w.r.t. the number of the clusters. This is consistent with the observation in SwAV.
>
> | # of clusters | 300 | 1000 | 3000 | 10000 | 30000 |
> |:---:|:---:|:---:|:---:|:---:|:---:|
> | Linear Top-1 | 67.7 | 69 | 69.3 | 69.5 | 69.5 |
> | k-NN (100%) | 58.9 | 60.5 | 61.6 | 61.7 | 61.7 |
> | k-NN (10%) | 49.5 | 51.9 | 53.3 | 53.3 | 53.3 |
> | k-NN (1%) | 36.6 | 39.7 | 41.0 | 41.0 | 41.0 |

---

### Official Review · Reviewer_UDR3 · 2022-07-11

**Rating:** 7
**Confidence:** 4
**Soundness:** 4 excellent
**Presentation:** 3 good
**Contribution:** 3 good

**Summary:**

The paper proposed a novel clustering-based self-supervised learning (SSL) method, MIRA. Motivated by the infomax principle, the paper constructs the SSL problem as an optimization problem that maximizes the mutual information between pseudo labels and data while taking the predictions of the model into consideration. To solve the online-clustering problem, the paper proposed a fixed-point iteration that takes only a few iterations to produce desirable pseudo-labels. Linear evaluation, KNN, and semi-supervised results on ImageNet verify the effectiveness of the method.

**Questions:**

1. As mentioned in the paper, MIRA converges faster than SwAV, and introduces MI regularization for better pseudo labeling. I guess MIRA will achieve better performance with 800 epochs of training. Or, will the method suffers a performance drop like TWIST from 400 epochs to 800 epochs?
2.  In Tab.8, it seems the reported results are trained without multi-crops, will the phenomenon still be the same when multi-crops are applied?
3. Will MIRA achieve desirable performance on downstream detection/segmentation tasks?

**Ethics Review Area:**

["I don’t know"]

**Limitations:**

Yes.

**Strengths And Weaknesses:**

Strengths:
1. The paper is well-motivated, and the loss design is simple and neat.
2. The paper is well-organized.
3. The authors provide plenty of experiments to demonstrate the effectiveness of the method. The method achieves state-of-the-art on several evaluations.

Weakness:
1. The paper did not mention the performance of MIRA on downstream detection/segmentation tasks. It would be better if the method can also achieve better performance on detection benchmarks as detection is one of the main downstream tasks for computer vision.
2. The paper did not provide the linear evaluation results with 800 epochs training to clarify the effectiveness of introducing Mutual information regularization. As Proposition 1 provides faster convergence, an 800-epochs result can further verify the improvement of pseudo-labels using MI regularized cluster assignment in Eq.3.

---

> ### Author Response · Authors · 2022-08-02
> **Response to reviewer UDR3**
>
> We appreciate the reviewer for emphasizing our proposed method as "well-motivated, simple, and neat" and highlighting the extensive experiments featured in this paper.
>
> > Q1. 800ep training results with MIRA.
> >
> > Q2. Multi-crop augmentation applied to a smaller batch size in Tab. 8?
>
> &rarr; Unfortunately, the experiments about Q1/Q2 cannot be finished in the *Author Rebuttal* phase due to limited time. We will answer Q1/Q2 with experimental results in the discussion period as soon as the experiments are done, in less than three days.
>
> > Q3. Will MIRA achieves desirable performance on downstream detection/segmentation tasks?
>
> &rarr; We test our method on detection/segmentation of the COCO 2017 dataset with Masked R-CNN, R50-C4 on a 2x scheduled setting. We use the configuration from the MoCo official implementation.  MIRA performs better than the supervised baseline and is comparable to MoCo; is not as desirable as in the classification tasks. The reviewer mBbA pointed out, methods that consider local or pixel-wise information show superior performance in these tasks. Incorporating such formulation into clustering-based approaches or MIRA seems to be an important future direction.
>
> |      | $\text{AP}^\text{bb}$ | $\text{AP}^\text{bb}_\text{50}$ | $\text{AP}^\text{bb}_\text{75}$ | $\text{AP}^\text{mk}$ | $\text{AP}^\text{mk}_{50}$ | $\text{AP}^\text{mk}_{75}$ |
> |------|:-----:|:--------:|:--------:|:-----:|:--------:|:--------:|
> |  Sup |   40  |   59.9   |   43.1   |  34.7 |   56.5   |   36.9   |
> | MoCo |  40.7 |   60.5   |   44.1   |  35.4 |   57.3   |   37.6   |
> | MIRA |  40.6 |    61    |   44.1   |  35.3 |   57.2   |   37.3   |

---

> > ### Author Response · Authors · 2022-08-05
> > **Response to reviewer UDR3, Q1/Q2**
> >
> > We complete the experiments about Q1 and Q2. Here is our resposnes for them.
> >
> > > Q1. 800ep training results with MIRA.
> >
> > &rarr; We report MIRA 800 epochs training results on the linear and k-NN evaluations.
> >
> > | Epochs | Top-1 | k-NN (100%) | k-NN (10%) | k-NN (1%) |
> > |:---:|:---:|:---:|:---:|:---:|
> > | 400ep | 75.5 | 68.7 | 60.7 | 47.8 |
> > | 800ep | 75.4 | 68.8 | 61.1 | 48.2 |
> >
> > MIRA trained for 800 epochs shows a similar performance to the one trained for 400 epochs; it shows +0.1~+0.4% improved k-NN evaluation accuracy and -0.1% linear evaluation accuracy. While the performance change depends on the evaluation protocols, it seems hard to claim neither MIRA is further improved nor it suffers from a performance drop. We just note that the models trained for 400 and 800 epochs with MIRA achieve strong performance on both linear and k-NN evaluation.
> >
> >
> > > Q2. Multi-crop augmentation applied to a smaller batch size in Tab. 8?
> >
> > &rarr; In Table 8, we report the linear evaluation performance of MIRA for 100 epochs training with a smaller batch size of 512; showing that MIRA performs relatively well with a reduced batch size. We further evaluate the performance of MIRA with multi-crop and smaller batch sizes below.
> >
> > | Method | Multi-crop | Batch size | Top-1 | Top-5 |
> > |:---:|:---:|:---:|:---:|:---:|
> > | SwAV | O | 4096 | 72.1 | - |
> > | MIRA | O | 512 | 72.9 | 91.3 |
> > | MIRA | O | 4096 | 73.5 | 91.5 |
> >
> > With multi-crops, MIRA with a batch size of 512 achieves 72.9% accuracy which outperforms the 72.1% accuracy of SwAV with a batch size of 4096. The result is consistent with Table 8 (where no multi-crops are applied) that MIRA shows robust performance with smaller batch size.

---

### Official Review · Reviewer_mBbA · 2022-07-12

**Rating:** 7
**Confidence:** 4
**Soundness:** 4 excellent
**Presentation:** 3 good
**Contribution:** 3 good

**Summary:**

This paper proposes Mutual Information Regularized Assignment (MIRA) which is a pseudo-labelling algorithm for unsupervised representation learning inspired by information maximization. The pseudo-labelling procedure is formulated as an optimization problem solved via a fixed-point iteration method that maximize the mutual information between the label and data while being close to a given model probability. The proposed methodology has been claimed to be converging relatively faster and to be achieving state-of-the-art performance on various downstream tasks such as linear evaluation and transfer learning. In general, the paper is well written and it is easily understandable, and I haven't found any typos and grammatical mistakes.

**Questions:**

(1) What will happen if the predicted pseudo labels become unbalanced in an iteration? What is the guarantee that the predicted pseudo labels be reasonably balanced in the update step?

(2) Mutual information (MI) maximization has been widely used for representation learning, but what is the motivation of using MI for having pseudo labels and then using those pseudo labels for representation learning whereas one can directly use MI maximization for representation learning?


**Limitations:**

The authors have mentioned that the time complexity of the proposed MIRA model can be a limitation of the work, which I think is normal for self-supervised learning models. Another limitation could be the consideration of only the global information and not considering any kind of local information, which will make this method only applicable for global task, such as classification.


**Strengths And Weaknesses:**

**Strengths**

(1) The paper is theoretically well grounded. The method can obtain an approximate solution of equation (3) by utilizing a fixed point iteration technique which has theoretical guarantee of convergence. In a nutshell, the proposed technique used optimal transport based pseudo labelling by maximizing the mutual information that avoids using hard artificial constraints. Altogether the idea looks very cool.

(2) On different downstream tasks, the proposed model has consistently outperformed the existing methodologies. The method has been thoroughly ablated and different components of the method are well justified.

**Weaknesses**

(1) In this method the optimal transport based pseudo labelling by maximizing the mutual information is the main novel component, which I find is less ablated. Also justification behind a two step pipeline (1. pseudo label prediction, 2. classification using those pseudo labels) is not clear. Representation can be learnt by directly maximizing MI.

(2) It is little difficult to follow the update rule in equation (5) from the unique optimal point found in equation (4). To this end, more explanation and references will be helpful to the reader.

(3) The paper has shown experiments on linear evaluation, semi-supervised learning, k-nn evaluation, transfer learning etc,, which are basically image classification experiments. More diverse experiments (such as object detection, semantic segmentation etc) are missing, which is not completely validating the success of the proposed MIRA model. Note that most of the existing self-supervised learning models do consider diverse tasks.

---

> ### Author Response · Authors · 2022-08-02
> **Response to reviewer mBbA - 1**
>
> Thank you for your extensive review. We are highly grateful that you praised our proposed method and empirical experiments. We are happy to give the responses to questions (Q) and weaknesses (W) below and hope they resolve your concerns.
>
> > Q1. What will happen if the predicted pseudo labels become unbalanced in an iteration? What is the guarantee that the predicted pseudo labels be reasonably balanced in the update step?
>
> &rarr; Our method, MIRA, finds mutual information (MI) maximized pseudo-labels through the optimization problem (Eq. 3). When model predictions (= predicted pseudo-labels) are extremely unbalanced (close to collapsing), the MI term of (Eq. 3) will be low. Through MI regularization, MIRA finds MI-maximizing points around the model predictions. Hence, MIRA will not choose collapsed (or extremely unbalanced) pseudo-labels; it is more likely to choose reasonably balanced ones to maximize the MI.
>
> Meanwhile, we can't guarantee that MIRA will escape from any collapsed point. There are some collapsed states that MIRA can't escape. For example, when the model outputs constant representation regardless of input, the effect of mutual information will be diminished; the output of MIRA will be fixed in the collapsed representation. However, this rarely happens in a conventional training scheme that starts from a non-collapsed state.
>
> > Q2/W1-2. What is the motivation of using MI for having pseudo-labels and then using the pseudo-labels for representation learning?
>
> &rarr; This work deals with clustering-based (or pseudo-labeling-based) representation learning. The advantage of these approaches is that they can account for inter-data similarity; representations are encouraged to encode the semantic structure of data. For instance, data representations that are assigned to the same cluster pull each other; in contrast, conventional NCE-based approaches perform instance-wise discrimination by pulling each other. Finding the similarity between the data seems important, especially for downstream tasks of classification; the empirical results in [A] show strong results of clustering-based approaches on classification tasks, e.g., transfer learning.
>
> > W1-1. Limited ablation study about pseudo-labeling by MI maximization.
>
> &rarr; To address the review's concern, we report more ablation studies about pseudo-labeling with MIRA. Reviewer E9n2 and y6Mg also suggest more ablation studies on pseudo-labeling, especially about (1) the number of clusters and (2) the different steps of iterations, respectively. We describe the results below:
> * The number of clusters
>
> | # of clusters | 300 | 1000 | 3000 | 10000 | 30000 |
> |:---:|:---:|:---:|:---:|:---:|:---:|
> | Linear Top-1 | 67.7 | 69 | 69.3 | 69.5 | 69.5 |
> | k-NN (100%) | 58.9 | 60.5 | 61.6 | 61.7 | 61.7 |
> | k-NN (10%) | 49.5 | 51.9 | 53.3 | 53.3 | 53.3 |
> | k-NN (1%) | 36.6 | 39.7 | 41 | 41 | 41 |
>
> We train MIRA on ImageNet-1k for 100 epochs without multi-crop augmentations while varying the number (#) of clusters. When the number of clusters is sufficiently large enough (>=3000), we observe no particular gain w.r.t. the number of the clusters.
>
> * The number of the fixed point iterations (*modified*)
>
> | # of iterations | 1it | 3it | 30it |
> |---|:---:|:---:|:---:|
> | MIRA 100ep | 69.5 | 69.5 | 69.3 |
>
> In the table, the convergence to the fixed point doesn't seem really important for downstream performance; it seems possible to choose a small number of iterations in practice. ~~However it is uncertain if such results hold when loss becomes sufficiently small as it approaches convergence. We will later update the results on a longer epoch during the discussion period~~. We update the linear evaluation results with the longer epochs (400) training below:
>
> | # of iterations | 1it | 3it | 30it |
> |:---:|:---:|:---:|:---:|
> | MIRA, 100ep | 69.5 | 69.5 | 69.3 |
> | MIRA, 400ep | 72.7 | 72.6 | 72.9 |
>
> In the 100 epochs training, the models with a smaller number of fixed point iterations, 1it and 3it, perform slightly better (+0.2%); while in 400 epochs training, the model with more iterations, 30it, performs better (+0.2~0.3%). As a result, the convergence to the fixed point is not the major factor for learning; however, it seems reasonable to choose a sufficiently large number of fixed point iterations in longer epoch training.

---

> > ### Author Response · Authors · 2022-08-02
> > **Response to reviewer mBbA - 2**
> >
> > > W2. More explanation and references will be helpful to follow the update rule in (Eq.5) from optimality condition (Eq.4).
> >
> > &rarr; Indeed, we skip the derivation from (Eq. 4) to (Eq. 5) in the main paper; the detailed derivation is only described in line 377 of the Appendix. We again note the derivation of (Eq. 5) from (Eq. 4) here:
> >
> > *By substituting the necessary and sufficient condition (Eq. 4) into* $\overline{w}\_j=\frac{1}{B}\sum\_{i=1}^B w\_{ij}$, *we get the necessary and sufficient condition with* $\overline{w^*}$.
> >
> > $$ \overline{w^\*}\_j = \frac{1}{B}\sum\_{i=1}^B w^\*_{ij}
> >     = \frac{1}{B}\sum\_{i=1}^B \frac{\overline{w^\*}\_j^{-\frac{\beta}{1-\beta}} p\_{ij}^{\frac{1}{1-\beta}}}{\sum\_{k=1}^K \overline{w^\*}\_k^{-\frac{\beta}{1-\beta}} p\_{ik}^{\frac{1}{1-\beta}}} = \overline{w^\*}\_j^{-\frac{\beta}{1-\beta}} \frac{1}{B}\sum\_{i=1}^B  \frac{ p\_{ij}^{\frac{1}{1-\beta}}}{\sum\_{k=1}^K \overline{w^\*}\_k^{-\frac{\beta}{1-\beta}} p\_{ik}^{\frac{1}{1-\beta}}} \Leftrightarrow \overline{w^\*}\_j = \Bigg[ \frac{1}{B}\sum\_{i=1}^B  \frac{ p\_{ij}^{\frac{1}{1-\beta}}}{\sum\_{k=1}^K \overline{w^\*}\_k^{-\frac{\beta}{1-\beta}} p\_{ik}^{\frac{1}{1-\beta}}} \Bigg]^{1 - \beta}. $$
> >
> > *And the RHS of the last equation becomes the update rule (Eq. 5).*
> >
> > We will incorporate this derivation into the main paper in the revised version.
> >
> > > W3. More diverse downstream tasks (detection/segmentation).
> >
> > &rarr; We test our method on detection/segmentation of the COCO 2017 dataset with Masked R-CNN, R50-C4 on a 2x scheduled setting. We use the configuration from the MoCo official implementation.  MIRA performs better than the supervised baseline and is comparable to MoCo; is not as dominating as in the classification tasks. As the reviewer pointed out, methods that consider local or pixel-wise information show superior performance in these tasks. Incorporating such formulation into clustering-based approaches or MIRA seems to be an important future direction.
> >
> > |      | $\text{AP}^\text{bb}$ | $\text{AP}^\text{bb}_\text{50}$ | $\text{AP}^\text{bb}_\text{75}$ | $\text{AP}^\text{mk}$ | $\text{AP}^\text{mk}_{50}$ | $\text{AP}^\text{mk}_{75}$ |
> > |------|:-----:|:--------:|:--------:|:-----:|:--------:|:--------:|
> > |  Sup |   40  |   59.9   |   43.1   |  34.7 |   56.5   |   36.9   |
> > | MoCo |  40.7 |   60.5   |   44.1   |  35.4 |   57.3   |   37.6   |
> > | MIRA |  40.6 |    61    |   44.1   |  35.3 |   57.2   |   37.3   |
> >
> >
> > [A] L. Ericsson, H. Gouk, and T. M. Hospedales. How well do self-supervised models transfer? In CVPR, 2021.

---

> > > ### Comment · Reviewer_mBbA · 2022-08-08
> > > **Thanks for the response and additional experimental results**
> > >
> > > I thank the authors for answering my comments and providing additional experimental results, which have been useful for improving my overall understanding of the work. I would expect the authors to include the results on the detection and segmentation downstream tasks in the paper and discuss the limitation of the work clearly. Furthermore, I think for considering representations encouraging to encode semantic structure of data, following few works [a,b] could be discussed and compared in the related works section. Anyway, I remain absolutely positive about this paper and have updated my review rating to "Accept".
> > >
> > > [a] Wang et al., Unsupervised Representation Learning by Invariance Propagation, NeurIPS, 2020.
> > >
> > > [b] Dwibedi et al., With a Little Help from My Friends: Nearest-Neighbor Contrastive Learning of Visual Representations, ICCV, 2021.

---

> > > > ### Author Response · Authors · 2022-08-09
> > > > **Thanks for the reply!**
> > > >
> > > > We are happy to hear that our responses have been helpful for the understanding of our work. We will incorporate the additional results and feedback, including the limitations, into the next version. For the recommending papers [a, b], we will add discussion on the papers in the revised version.
> > > >
> > > > Thank you!
> > > >
> > > > Authors.

---

### Author Response · Authors · 2022-08-02
**General response**

We thank the reviewers for their time and effort to provide constructive reviews. We appreciate the encouraging remarks on the paper, including novelty (UDR3, E9n2, y6Mg), well-defined formulation (mBbA), neat organization (UDR3), and extensive experimental evaluations (UDR3, E9n2).

Due to limited time and resources, we are not able to address all the comments in the Author Rebuttal period, some experiments are currently in progress. For these experiments, we are going to update our comments as soon as they are finished; we will further revise the paper after updating the comments.

---

> ### Author Response · Authors · 2022-08-07
> **Revision uploaded**
>
> We updated the paper with the following modifications:
> * We briefly explain our solving strategy in the fourth paragraph of the introduction.
> * We add more details on the 3.2 solving strategy, especially about the derivation of the necessary and sufficient conditions and the fixed point iteration.
> * We fix some typos.

---

### Meta-Review · Area_Chair_k2PP · 2022-08-25

**Recommendation:** Accept
**Confidence:** Certain

**Metareview:**

This paper proposes a pseudo-labelling algorithm for unsupervised representation learning inspired by information maximization.

The reviewers found  that the proposed method is theoretically well grounded and that the authors provide extensive experimentations to demonstrate the validity of their approach. I agree with those conclusion after reading the paper. I therefore support acceptance.

**Award:**

No

---

### Decision · Program_Chairs · 2022-09-14

Accept